# Germ granule localization of nematode Argonaute WAGO-4 ensures fidelity in small RNA loading

Stela Jelenic [1,2✉], Mathias S Renaud[3], Samantha Del Borrello [3], Joseph Gokcezade[1], Janos Bindics [1,4,5], Lisa Frasz[1,6], Philipp Czermak[1,7], Peter Duchek[1] & Julie M Claycomb [3]

## Abstract

**Germ granules are liquid-like condensates that regulate small RNA pathways and gene expression, ensuring genome stability and fertility in animals. In *C. elegans*, several Argonaute proteins, central players of small RNA pathways, localize to germ granules, yet the functional significance of this spatial enrichment remains unclear. Here, we disrupted the localization of the Argonaute WAGO-4 to germ granules by introducing targeted mutations in the FG repeats of Vasa-like GLH proteins. These mutations did not disrupt overall germ granule architecture but significantly reduced WAGO-4 partitioning, leading to its predominant localization in the cytoplasm. Functional analyses revealed that this mislocalization partially compromised WAGO-4 activity, resulting in reduced WAGO-4 binding of small RNAs targeting specific genes, particularly those not co-regulated by the Argonaute CSR-1. This selective effect highlights the importance of WAGO-4's spatial localization for efficient small RNA loading and gene regulation. Our findings demonstrate that germ granules serve as specialized compartments that fine-tune Argonaute function, emphasizing the role of phase-separated condensates in modulating RNA pathways and gene regulatory networks.**

**Keywords** Argonautes; Biomolecular Condensates; *C. elegans* Germline; Germ Granules; Small RNA Pathways
**Subject Category** RNA Biology

## Introduction

In many organisms, the germline contains liquid-like condensates that play essential roles in germ cell maintenance, transgenerational epigenetic inheritance, and fertility (Voronina et al, 2011; Trcek and Lehmann, 2019; Tian et al, 2020; Cecere, 2021; Dodson and Kennedy, 2020; Sundby et al, 2021). These condensates, known as germ granules, regulate small RNA (sRNA) pathways and gene expression, ensuring genome stability and reproductive success. Germ granules form via liquid–liquid phase separation (LLPS), a process that enables the selective recruitment and concentration of proteins and RNAs (Banani et al, 2017). Among their key components are Argonaute (Ago) proteins, which are central to sRNA-mediated gene regulation (Dodson and Kennedy, 2020; Cecere, 2021; Sundby et al, 2021; Pamula and Lehmann, 2024).

The *C. elegans* germline contains multiple, functionally connected condensates, including P granules, Z granules, Mutator foci, SIMR foci, and E granules (Phillips et al, 2012; Wan et al, 2018; Uebel et al, 2023; Manage et al, 2020; Chen et al, 2024). These structures coordinate different aspects of sRNA pathways by facilitating sRNA biogenesis, modification, and Ago loading (Phillips and Updike, 2022; Sundby et al, 2021). P granules, adjacent to the nuclear envelope, contain multiple Argonautes, including CSR-1, PRG-1, and WAGO-1, which regulate germline gene expression (Batista et al, 2008; Claycomb et al, 2009; Gu et al, 2009). Z granules, which localize next to P granules, play a key role in secondary sRNA amplification and transgenerational gene silencing (Ishidate et al, 2018; Wan et al, 2018; Ouyang et al, 2022). These granules contain WAGO-4, an Argonaute that also localizes to P granules and functions in germline sRNA pathways (Wan et al, 2018; Seroussi et al, 2023). WAGO-4 is required for the inheritance of sRNAs that target genes silenced by exogenous RNAi and regulates nearly 5,000 endogenous targets, many of which overlap with CSR-1 (Xu et al, 2018; Wan et al, 2018; Charlesworth et al, 2021; Seroussi et al, 2023). While WAGO-4 and CSR-1 are proposed to have antagonistic roles in germline gene regulation, their functional relationship remains incompletely understood (Wedeles et al, 2013; Seth et al, 2013; Xu et al, 2018).

Although germ granule localization is a conserved feature of many Argonautes, its precise functional significance remains unclear. Studies in diverse systems suggest that condensates may enhance sRNA-mediated gene silencing by promoting Ago loading, stability, or activity. However, assessing the role of Ago enrichment in vivo is challenging because most functional studies rely on genetic modifications of Ago proteins or major perturbations of germ granules, which affect multiple pathways (Aoki et al, 2021;

[1]Institute of Molecular Biotechnology of the Austrian Academy of Sciences (IMBA), Vienna BioCenter (VBC), Dr. Bohr-Gasse 3, Vienna 1030, Austria. [2]Vienna BioCenter PhD Program, A Doctoral School of the University of Vienna and the Medical University of Vienna, Vienna, Austria. [3]Department of Molecular Genetics, University of Toronto, Toronto, ON M5G 1M1, Canada. [4]Present address: BOKU University, Universität für Bodenkultur Wien, Muthgasse 18, Vienna 1190, Austria. [5]Present address: Austrian Centre of Industrial Biotechnology (acib GmbH), Krenngasse 37/2, Graz 8010, Austria. [6]Present address: Research Institute of Molecular Pathology (IMP), Vienna BioCenter (VBC), Campus-Vienna-Biocenter 1, Vienna 1030, Austria. [7]Present address: BioNTech R&D Austria, Helmut-Qualtinger-Gasse 2, Vienna 1030, Austria. ✉E-mail: stela.jelenic@imba.oeaw.ac.at

Dai et al, 2022; Chen et al, 2022). For instance, in *C. elegans*, disrupting P granules via null mutations in *pgl* genes (which encode RNA-binding proteins localized to germ granules) or the *glh* genes (encoding Vasa-like helicases) mislocalizes multiple Agos but also alters granule integrity, making it difficult to disentangle specific effects on Ago function. Similarly, mutations affecting proteins that recruit Agos to germ granules, such as HRDE-2 and PEI-1/PEI-2, impair sRNA pathways (Schreier et al, 2022; Chen and Phillips, 2024), but whether this reflects disrupted spatial localization or the loss of specific interactions remains unclear. These observations suggest that granule-associated localization is functionally relevant, yet its direct impact on Ago function remains unresolved.

To address this, we leveraged a system that selectively disrupts WAGO-4 partitioning into germ granules without broadly affecting granule integrity. In *C. elegans*, the Vasa-like GLH proteins (GLH-1, GLH-2, and GLH-4) are constitutive P granule components that contain FG repeats—motifs typically associated with nuclear pore complex (NPC) function (Updike et al, 2011; Shinkai et al, 2021). FG repeats mediate weak and transient interactions in NPCs, and we hypothesized that they might similarly contribute to condensate composition (Ribbeck and Görlich, 2002; Patel et al, 2007; Updike et al, 2011). By introducing targeted mutations that perturb nearly all FG repeats in GLH-1, GLH-2, and GLH-4, we disrupted WAGO-4 recruitment to germ granules while preserving granule structure. This approach allows us to directly assess the importance of WAGO-4's spatial enrichment in regulating gene expression.

Our findings reveal that WAGO-4 localization within germ granules is critical for its function. In animals bearing mutations in the FG domains of GLH proteins, WAGO-4 mislocalization to the cytoplasm correlates with defects in sRNA loading and target gene regulation. These effects are particularly pronounced for genes that are not co-targeted by CSR-1, suggesting that spatial compartmentalization enhances specific aspects of WAGO-4 function. This work demonstrates that phase-separated compartments in the germline do more than passively concentrate RNA regulatory factors—they actively modulate Ago function, fine-tuning sRNA pathways to maintain proper gene regulation.

## Results

### FG repeats of GLH proteins contribute to fecundity at elevated temperatures

To investigate the role of FG repeats in the germ granules of *C. elegans*, we introduced mutations in the FG repeats of GLH-1, GLH-2, and GLH-4 using CRISPR-Cas to target the native loci of the genes (Fig. 1A). GLH-3 was not included in this analysis, as it does not contain FG repeats. In GLH-1 and GLH-4, phenylalanine-to-alanine ($F \rightarrow A$) substitutions were introduced at phenylalanine-glycine (FG) dipeptides. Due to technical limitations, a large deletion covering most of the FG repeats was made in GLH-2, leaving the zinc finger domains intact ($\Delta 17\text{-}244$, i.e., $\Delta FG$). The number of FG or GF dipeptides present and mutated in each gene is listed in Fig. EV1A, and the protein sequences of FG mutants are provided in Table EV1.

Given the established role of GLH proteins in *C. elegans* fertility and fecundity (Spike et al, 2008; Kuznick et al, 2000), we assessed the reproductive outcomes of mutants in FG domains of GLH proteins (hereafter referred to as FG mutants). At the permissive temperature of 20 °C, no significant difference in progeny count was observed between wild-type (WT) and FG mutants (Fig. EV1B). As many *C. elegans* germline mutants exhibit temperature-sensitive phenotypes, we tested the reproductive outcomes at an elevated temperature (26 °C). The mutation of GLH-1 FG repeats alone did not significantly affect progeny count compared to WT. However, double (GLH-1 and GLH-4) and triple (GLH-1, GLH-2, and GLH-4) FG mutants displayed significantly reduced progeny counts with notable variability (Fig. 1B). For simplicity, we will hereafter refer to animals with FG-domain mutations in GLH-1 as 1×FG; with simultaneous mutations in GLH-1 and GLH-4 as 2×FG; and with simultaneous mutations in GLH-1, GLH-2, and GLH-4 as 3×FG mutants (Fig. 1A). Depending on the assay, 2×FG and 3×FG mutants showed either a more than 50% reduction in progeny or near-complete sterility (Fig. EV1C). We attribute this variability to slight differences in environmental conditions across assays. Nonetheless, the reduction in progeny for 2×FG and 3×FG mutants compared to WT was consistent and reproducible. As 26 °C is a high temperature for *C. elegans*, we also tested the fecundity of FG mutants at 25 °C and observed that both 2×FG and 3×FG mutants produced significantly fewer progeny compared to wild-type animals (Fig. EV1D). Although the phenotype was already apparent at 25 °C, we conducted subsequent experiments at 26 °C because (a) the phenotype was more pronounced at this temperature, and (b) 26 °C has been commonly used in the literature to assess fertility in *glh* mutants (e.g., Marnik et al, 2019; Spike et al, 2008). We tested all the single FG mutants (*glh-1(F → A), glh-2(ΔFG),* and *glh-4(F → A))* to identify the contribution of each GLH protein FG domain to the observed fecundity phenotype at 26 °C. Our results showed that all single FG mutants exhibited similar progeny numbers to the N2 control (Fig. EV1E), indicating that mutations in multiple FG domains are required to observe the phenotype.

GLH proteins are not the only germ granule components containing FG repeats; other proteins, such as DDX-19 and RDE-12, also possess FG domains. The role of FG repeats in RDE-12 has been reported (Yang et al, 2014). Introducing an FG mutation in DDX-19 in addition to the 3×FG mutant (resulting in the 4×FG mutant) did not further exacerbate the phenotype (Fig. EV1F,G). Therefore, we focused on FG repeats in GLH paralogs, particularly the 3×FG mutant, which harbors mutations in all three FG-repeat-containing GLH proteins.

Several lines of evidence suggest that the FG mutations we introduced are not loss-of-function mutations: (1) 2×FG and 3×FG mutants displayed WT fecundity at the permissive temperature, unlike *glh-1/glh-4* double-null mutants, which are sterile at 20 °C (Spike et al, 2008); and (2) our in vitro ATPase assay showed no significant difference in ATPase activity between GLH-1 FG mutants and WT (Fig. EV1H). In addition, gonadal and germ cell morphology appeared normal in FG mutants, even when grown on elevated temperatures (26 °C), and did not display the severe defects, such as large, Y-shaped, or small germlines that were previously reported in *glh-1* or *glh-1/glh-4* null mutants. (Spike et al, 2008). However, we did observe unusual structures resembling giant cells in the uteri of FG mutants (Fig. 1C). Several observations suggest these structures are likely unfertilized oocytes: (1) they had passed through the spermatheca without forming a proper eggshell, as shown by FM4-64 dye staining (Fig. EV1I) (Singson, 2001; Olson

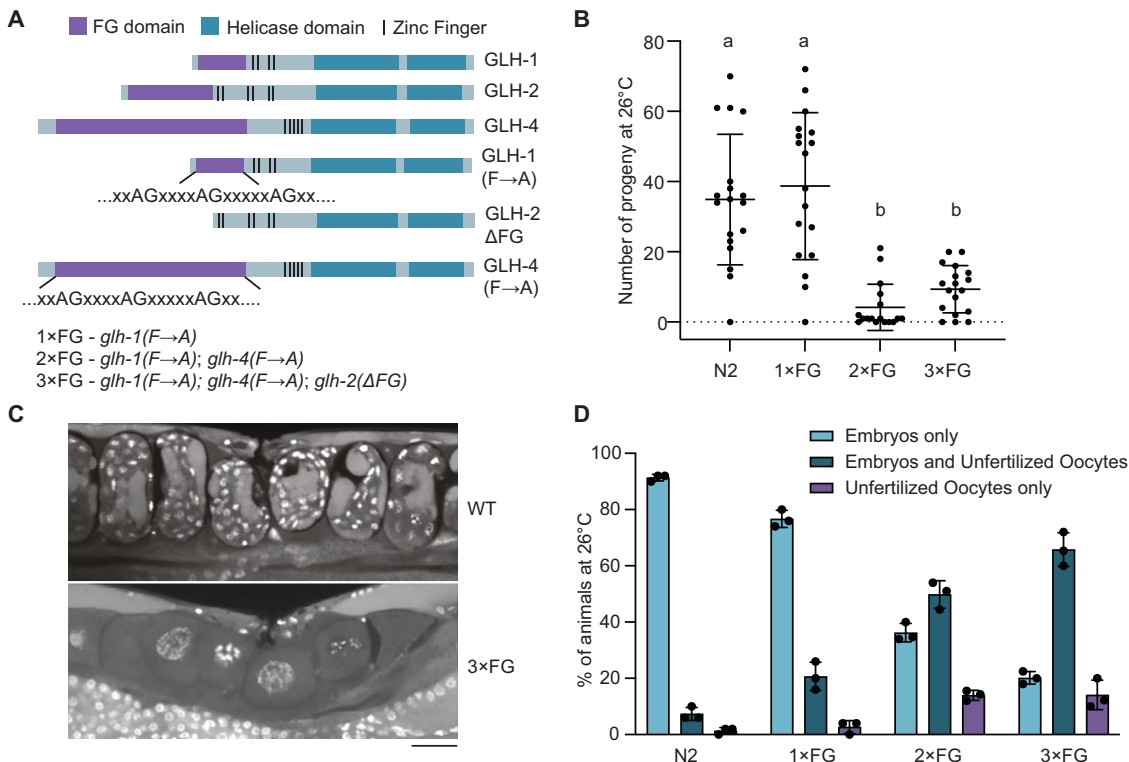

**Figure 1. FG repeats of GLH proteins contribute to fecundity at elevated temperatures.**

(A) Schematic representation of wild-type (WT) and mutated forms of GLH-1, GLH-2, and GLH-4 proteins. These mutations were introduced into the native loci of the respective genes. Individual *C. elegans* strains carry either single or multiple gene mutations. Abbreviated mutant names used in this study are listed at the bottom. (B) Number of progeny for worms of the indicated genotypes grown at 26 °C for one generation. Each dot represents the progeny count of a single worm (*n* = 17–18). Horizontal lines indicate mean values, and error bars represent standard deviation (SD). Statistical analysis was performed using the Kruskal–Wallis test, and the corresponding *p* values are reported in Dataset EV2. Groups sharing at least one letter above the plot are statistically indistinguishable (*P* < 0.05). (C) Fluorescence micrographs of fixed worm uteri stained with DAPI. Representative images show a WT worm containing stacked embryos (upper) and a 3×FG worm containing stacked unfertilized oocytes (lower). Worms were grown at 26 °C for one generation. Scale bar: 20 μm. (D) Quantification of unfertilized oocytes in the uteri of worms grown at 26 °C for one generation. Each worm was categorized into one of three groups: (1) containing exclusively embryos, (2) containing at least one embryo and at least one unfertilized oocyte, or (3) containing exclusively unfertilized oocytes. Data represent the average proportion of worms in each category from three independent experiments (*n* = 49.4 ± 1.4 worms per experiment). Dots represent values from individual experiments, and error bars indicate SD. Source data are available online for this figure.

et al, 2012); (2) they exhibited strong DAPI staining, indicating DNA accumulation from endomitotic replication without cytokinesis (Fig. 1C) (Singson, 2001); and (3) they were laid and morphologically matched the description of previously reported unfertilized oocytes (Singson, 2001).

At 20 °C, unfertilized oocytes were rare in both WT and FG mutants (Fig. EV1J). In contrast, at 26 °C, only ~10% of WT young gravid adults contained one or more unfertilized oocytes, while ~80% of 3×FG mutants exhibited this phenotype (Fig. 1D). The proportion of embryos to unfertilized oocytes varied, ranging from a single unfertilized oocyte alongside multiple embryos to the presence of multiple unfertilized oocytes with few or no embryos. The severity of the phenotype was highest in 3×FG mutants and did not worsen in the 4×FG mutants (Fig. EV1K). To determine whether this phenotype was due to sperm defects, we performed mating assays. 2×FG and 3×FG mutant hermaphrodites successfully mated with WT males, but fecundity was not restored to WT levels (Fig. EV1L), suggesting that the defect lies in oocyte quality.

In summary, our results indicate that FG-repeat mutations in GLH proteins impair fertility and fecundity in *C. elegans* at elevated temperatures, primarily manifested as an accumulation of unfertilized oocytes.

## FG-repeat mutations in GLH proteins preserve perinuclear germ granule localization but alter P granule composition

GLH proteins are constitutive components of P granules (Updike and Strome, 2010), making it essential to evaluate P granule formation and localization in FG mutants. Using live imaging with the P granule marker PGL-3::mCherry, we found that P granules were formed and properly localized to the perinuclear region in the adult germline of 3×FG mutants at both 20 °C and 26 °C (Fig. 2A). Quantitative analysis revealed a slight increase in the number of P granules but an approximately 20% reduction in total P granule volume (Fig. EV2A,B).

In the adult germline, germ granules comprise several subcompartments, including Z granules and Mutator foci, whose formation depends on P granules (Wan et al, 2018; Sundby et al, 2021; Singh et al, 2021). To assess the impact of FG mutations on

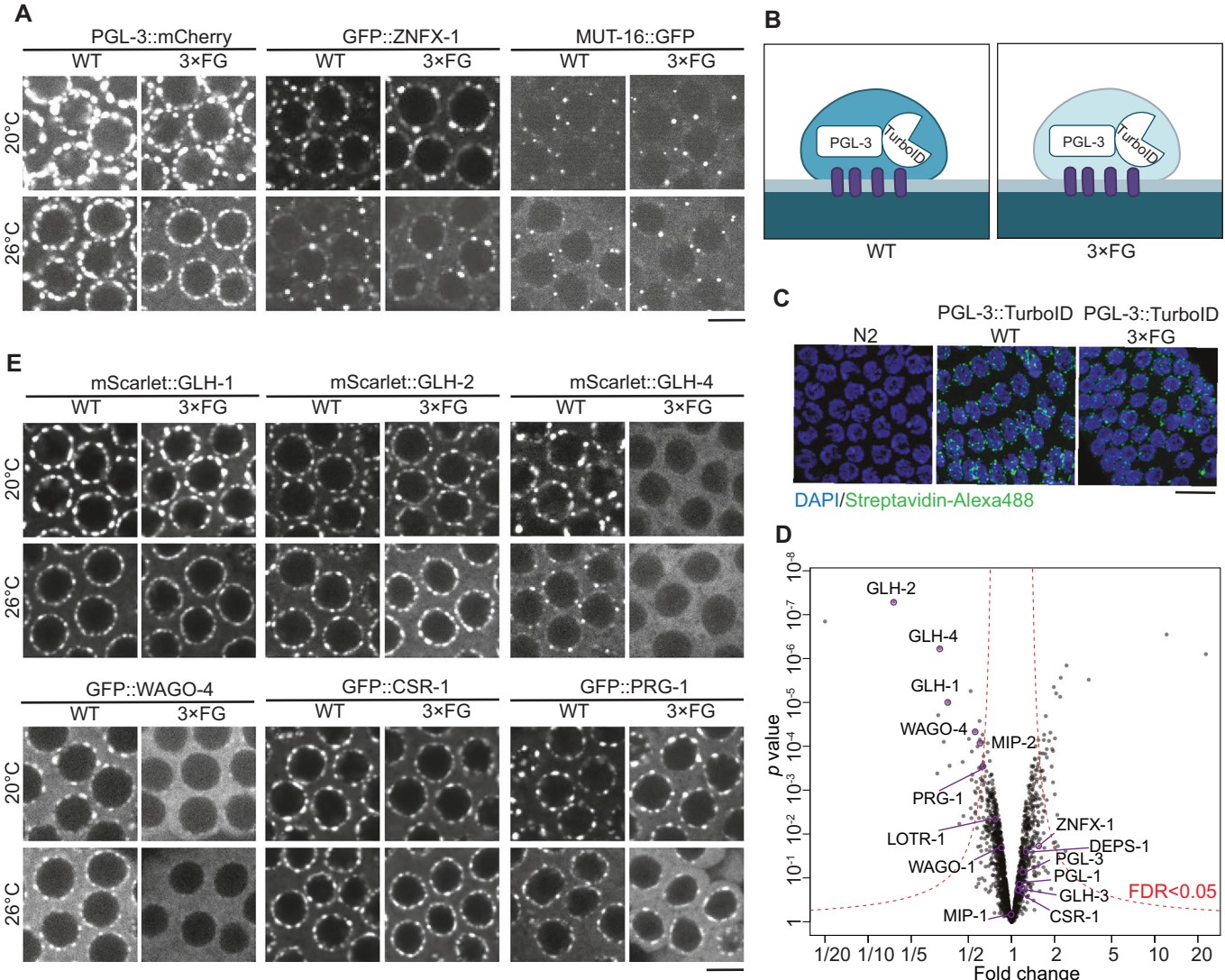

**Figure 2.** FG-repeat mutations in GLH proteins preserve perinuclear germ granule localization but alter P granule composition.

(A) Representative fluorescence micrographs of gonads (pachytene region) from live worms expressing single fluorescently labeled proteins: PGL-3::mCherry, RFP::ZNFX-1, and MUT-16::GFP. Micrographs show a single confocal slice for WT and 3×FG backgrounds at 20 °C and 26 °C. Scale bar: 5 µm. (B) Schematic of the proximity labeling experiment. Biotin ligase (TurboID) was fused to the C-terminus of PGL-3 and expressed in WT and 3×FG mutant backgrounds. Protein biotinylation levels served as a proxy for protein enrichment in P granules. (C) Fluorescent micrographs of fixed gonads showing maximal intensity projections after DAPI (blue) and Streptavidin-Alexa488 (green) staining. Streptavidin-Alexa488 signals appear as punctate perinuclear dots in strains expressing PGL-3::TurboID (both WT and 3×FG) but are absent in the negative control (N2). Scale bar: 10 µm. (D) Volcano plot showing fold enrichment of proteins after streptavidin pulldown in PGL-3::TurboID worms at 26 °C in the 3×FG vs. WT background, as determined by mass spectrometry ($n = 3$ biological replicates per genotype). Statistical significance was calculated using limma (Ritchie et al, 2015). Identified known P granule and Z granule proteins are highlighted. (E) Representative fluorescence micrographs of gonads (pachytene region) from live worms expressing different fluorescently labeled proteins: mScarlet::GLH-1, mScarlet::GLH-2, mScarlet::GLH-4, GFP::WAGO-4, GFP::CSR-1, and GFP::PRG-1. For each fluorescent protein, a single confocal slice is shown for worms with WT and 3×FG backgrounds grown at 20 °C and 26 °C. Scale bar: 5 µm.

these sub-compartments, we used fluorescent markers GFP::ZNFX-1 and MUT-16::GFP to visualize Z granules and Mutator foci, respectively. Both sub-compartments were formed and localized perinuclearly in 3×FG mutants at 20 and 26 °C (Fig. 2A). Interestingly, we observed a subtle change in the 3D morphology of Z granules, indicating increased mixing between P granules and Z granules (Fig. EV2C,D).

Although no obvious defects were detected in GLH protein function or germ granule organization in 3×FG mutants, the underlying cause of the fertility defects remained unclear. We hypothesized that FG repeats might play a role in establishing dynamic protein-protein interactions necessary for proper P granule composition. To test this hypothesis, we used a proximity labeling approach by tagging PGL-3 with TurboID via CRISPR-Cas at its native locus and expressing PGL-3::TurboID in both WT and 3×FG mutant backgrounds at both 20 °C and 26 °C (Fig. 2B). Immunofluorescence with Streptavidin-Alexa Fluor 488 confirmed that the proximity labeling construct was active and properly

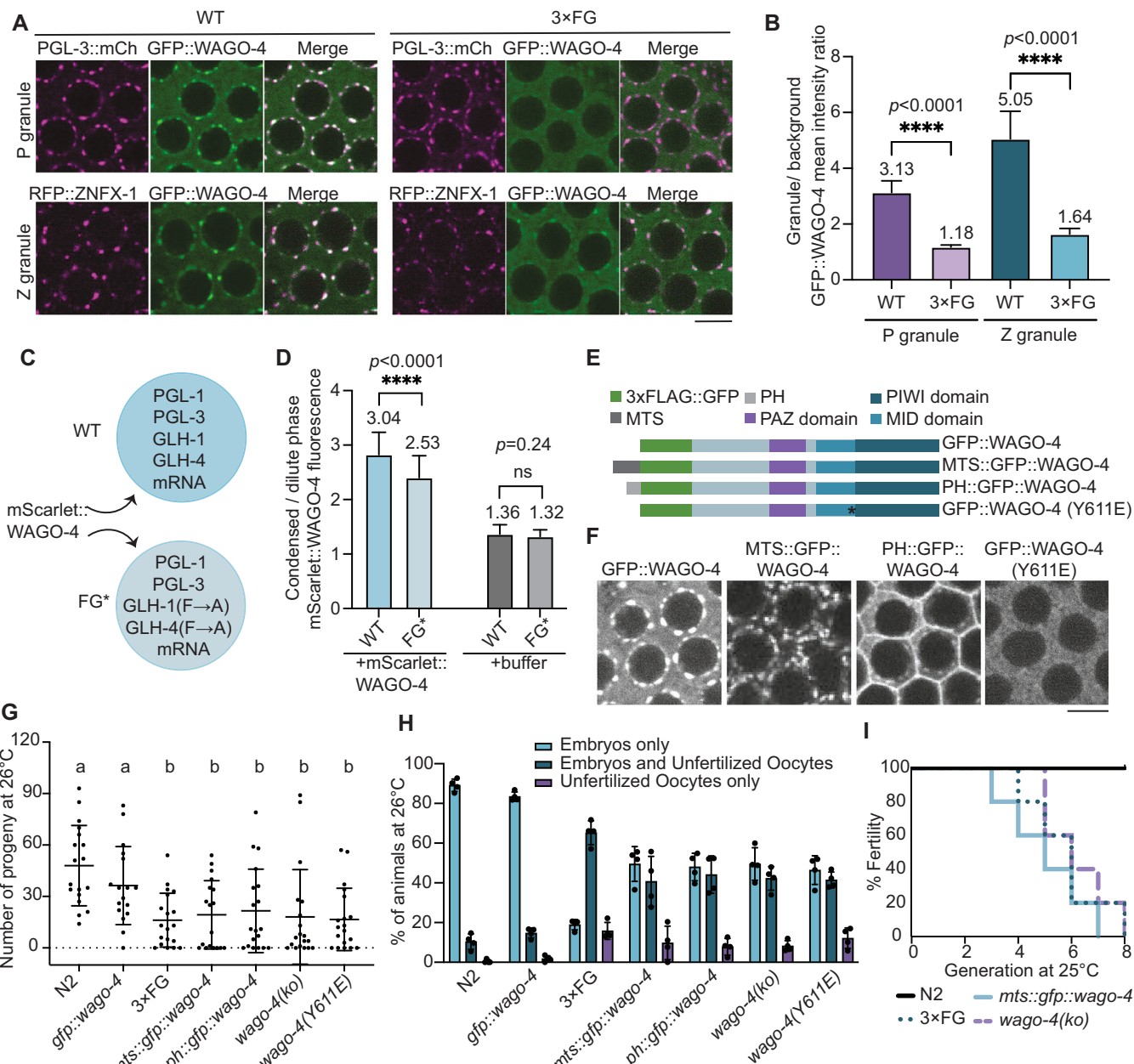

**localized to perinuclear puncta in both WT and 3×FG mutant worms (Fig. 2C).**

We assessed protein biotinylation levels in WT and 3×FG mutants using streptavidin pull-down followed by quantitative mass spectrometry analysis. The majority of germ granule proteins showed no significant change in biotinylation levels, with only a few known germ granule proteins exhibiting a significant reduction. At 20 °C, the reduction was restricted to GLH-1, GLH-2, and GLH-4—proteins with FG mutations (Fig. EV2E). However, at 26 °C, additional proteins such as WAGO-4, PRG-1, and MIP-2 also showed reduced biotinylation levels (Fig. 2D), suggesting decreased partitioning of these proteins into P granules.

We verified the localization of the candidate proteins by introducing fluorescent tags and using live imaging (Fig. 2E). Since

MIP-2 is expressed at low levels in the examined pachytene region of WT worms, we did not pursue further localization analysis for this protein (Cipriani et al, 2021). Both mScarlet::GLH-1 and mScarlet::GLH-2 were properly concentrated in perinuclear foci despite their reduced biotinylation signals. In contrast, mScarlet::GLH-4 showed a diffuse cytoplasmic signal with no detectable perinuclear puncta. Among the Argonaute proteins tested, GFP::WAGO-4 exhibited a significant reduction in perinuclear puncta signal intensity, while GFP::PRG-1 and GFP::CSR-1 remained concentrated in perinuclear foci. We estimated the perinuclear granule enrichment level for each protein by calculating the standard deviation of the signal intensity along the perinuclear region (Fig. EV2F,G).

Based on these data, we sought to identify the potential cause of the phenotype in 3×FG mutants. Our results suggest that the

◄ **Figure 3.   Reduced germ granule localization of WAGO-4 underlies FG mutant phenotype.**

(A) Representative fluorescent micrographs of gonads (pachytene region) from live worms co-expressing PGL-3::mCherry and GFP::WAGO-4 (upper row) or RFP::ZNFX-1 and GFP::WAGO-4 (lower row) in WT and 3×FG mutant backgrounds at 20 °C. For each condition, a single confocal slice is shown for the indicated proteins and merged channels. Scale bar: 5 µm. (B) Quantification of GFP::WAGO-4 mean intensity signal inside granules vs. the cytoplasm. Ratios were calculated from images of live worms expressing GFP::WAGO-4 together with PGL-3::mCherry (P granules, purple hues) or RFP::ZNFX-1 (Z granules, blue hues) in WT and 3×FG backgrounds. Average ratios with SD are plotted, and numbers above bars indicate average values. Analysis was performed on the pachytene region, covering approximately 35 nuclei for P granules (35.3 ± 2.8) and 30 nuclei for Z granules (29.7 ± 2.7) from three live worms grown at 20 °C. To ensure precise granule volume definition and accurate intensity ratio measurement, the granule vs. cytoplasm signal ratios were calculated for each of 54 confocal planes (0.1 µm). Statistical analysis was performed using a two-tailed Mann–Whitney test, *P* values indicated in the graph. Representative fluorescence micrographs are shown in (A). (C) Schematic representation of the in vitro phase separation experiment. Two types of condensates were formed to approximate minimal P granules. Each condensate contained recombinant purified PGL-1, PGL-3, GLH-1, and GLH-4 proteins from an insect cell expression system, along with commercial poly(A)+ mRNA. The first type of condensate contained WT GLH-1 and GLH-4, while the second contained GLH-1 and GLH-4 with F → A mutations in FG dipeptides (identical to those introduced in live worms, shown in Fig. 1A). To each condensate type, the same amount of worm-purified mScarlet::WAGO-4 was added. For negative controls, the same amount of WAGO-4 storage buffer was added to the condensates. (D) Graph representing the ratio of mScarlet::WAGO-4 signal intensity between the condensed and dilute phases for WT and FG-mutant condensates. Results from a single experiment are shown; five to six ROIs per image were defined inside and outside condensates, with six images analyzed for each condition (*n* = 32–36). Bars represent average values with SD and numbers above bars indicate average values. Statistical analysis was performed using a two-tailed Mann–Whitney test, *P* values indicated in the graph. A second independent experiment is shown in Fig. EV3B. mScarlet::WAGO-4 storage buffer was used as a negative control, and ratios were calculated in the same manner. (E) Schematic representation of WAGO-4 constructs expressed in worms. Fusions or point mutations were introduced at the native *wago-4* locus using CRISPR-Cas. Introduced fusions or WAGO-4 predicted domains, identified based on sequence alignment with human Ago2, are depicted in different colors, as indicated. (F) Representative fluorescence micrographs of gonads (pachytene region) from live worms expressing different fluorescently labeled WAGO-4 proteins: GFP::WAGO-4, MTS::GFP::WAGO-4, PH::GFP::WAGO-4, and GFP::WAGO-4(Y611E). For each expressed protein, a single confocal slice is shown for worms grown at 20 °C. Scale bar: 5 µm. (G) Graph representing the number of progeny for worms of the indicated genotypes grown at 26 °C for one generation. Each dot represents the number of progeny from a single worm (*n* = 17–19). Horizontal lines represent mean values, and error bars indicate SD. Statistical analysis was performed using the Kruskal–Wallis test, and the corresponding *P* values are reported in Dataset EV2. Groups sharing at least one letter above the plot are statistically indistinguishable (*P* < 0.05). (H) Quantification of unfertilized oocytes in the uteri of worms grown at 26 °C, categorized into three groups: (1) worms containing exclusively embryos, (2) worms containing at least one embryo and at least one unfertilized oocyte, and (3) worms containing exclusively unfertilized oocytes. Data represent the average proportion of worms in each category from four independent experiments (*n* = 52.8 ± 6.1 worms per experiment). Dots represent values from individual experiments and error bars indicate SD. (I) Mortal germline assay at 25 °C comparing the 3×FG mutant, MTS::GFP::WAGO-4-expressing strain, and *wago-4(ko)*, with N2 as a control. For each strain, five replicates with five P<sub>O</sub> worms were used. Source data are available online for this figure.

impact of FG-repeat mutations on germ granules in adult worms is subtle, with no evidence of their mislocalization or major compositional changes. However, we observed strong delocalization of WAGO-4 and GLH-4 from the germ granules. Since the *glh-4* null mutant does not exhibit a fertility phenotype (Spike et al, 2008), we focused our investigation on WAGO-4 localization. To evaluate whether differences in the protein levels of WAGO-4 (or GLH proteins) might contribute to the phenotype, we measured their abundance in the 3×FG mutants using mass spectrometry. Our analysis revealed that the levels of WAGO-4, as well as GLH-1, GLH-2, and GLH-4, in the 3×FG mutants are comparable to those in wild-type animals (Fig. EV2H). Considering these data, we hypothesized that even subtle changes in the FG mutants, such as the altered localization of WAGO-4, could contribute to the fertility defects observed under stress conditions.

## Reduced germ granule localization of WAGO-4 underlies FG mutant phenotype

WAGO-4 has been reported to localize to both P and Z granules (Seroussi et al, 2023; Weiser and Kim, 2019). In 3×FG mutants, we found that WAGO-4 is delocalized from both compartments (Fig. 3A,B). The granule-to-cytoplasm signal enrichment level for WAGO-4 dropped from 3.13 ± 0.40 in P granules and 5.05 ± 1.11 in Z granules in WT to 1.18 ± 0.03 and 1.63 ± 0.18, respectively, in 3×FG mutants. We next sought to test whether this reduced P granule partitioning could be observed in a minimal in vitro system. Using purified recombinant PGL-1, PGL-3, GLH-1, and GLH-4, along with commercial pure poly-A+ mRNA (Takara), we reconstituted minimal P granules by inducing liquid–liquid phase

separation (LLPS). Proteins were combined in stoichiometric ratios as described previously (Saha et al, 2016), and two types of condensates were formed: one containing WT GLH-1 and GLH-4 and another containing FG-mutated versions of these proteins (same mutations as in *C. elegans* strains, Fig. 1A). We then added equal amounts of mScarlet::WAGO-4 purified from worms to both condensates (Fig. 3C).

We measured the mScarlet signal ratio inside vs. outside the condensates as a proxy for WAGO-4's affinity to partition within the P granule-like phase. Even in this minimal system, we detected significantly lower levels of WAGO-4 partitioning in the FG-mutant condensates (Figs. 3D and EV3A). This result was reproducible in an independent experiment (Fig. EV3B). Although the difference in WAGO-4 partitioning between WT and FG-mutant condensates was less pronounced than in live worms, this suggests that FG repeats may directly contribute to WAGO-4 recruitment, while additional factors and active biological processes likely play a role in vivo.

To assess whether WAGO-4 delocalization could be an important contributor to the phenotype of FG mutants, we generated four *wago-4* mutants and tested their reproductive outcomes (Fig. 3E,F). Two mutants were designed to delocalize WAGO-4 artificially by fusing targeting sequences to the native *wago-4* gene: (1) a mitochondrial targeting sequence (MTS) to target WAGO-4 to mitochondria (Nargund et al, 2012) and (2) a PH domain to target WAGO-4 to the plasma membrane (Huelgas-Morales et al, 2020). MTS::GFP::WAGO-4 showed almost complete delocalization from the nuage, while PH::GFP::WAGO-4 retained some residual perinuclear punctate signal. In the third mutant, WAGO-4(Y611E) harbors a point mutation predicted to disrupt

sRNA binding (Rüdel et al, 2011) and exhibited a diffuse cytoplasmic signal. The fourth was a *wago-4(ko)* knockout mutant created by introducing a stop codon in the second exon of the *wago-4* gene using CRISPR-Cas. Absence of WAGO-4 protein expression in this strain was validated by mass spectrometry (Fig. EV3C).

Interestingly, all four *wago-4* mutants exhibited fertility phenotypes similar to each other and comparable to the 3×FG mutant. First, at 26 °C, all *wago-4* mutants showed significantly reduced progeny numbers compared to WT and had reproductive outputs similar to the 3×FG mutant (Fig. 3G). At 20 °C, however, fecundity remained at WT levels for all strains (Fig. EV3D). Second, the germline morphology of all *wago-4* mutants was similar to 3×FG mutants, with no obvious structural defects; however, unfertilized oocytes were frequently observed in the uteri of worms grown at 26 °C (Fig. 3H). At 20 °C, unfertilized oocytes were rarely detected across all strains (Fig. EV3E). Notably, the frequency of unfertilized oocytes at 26 °C was higher in 3×FG mutants compared to all *wago-4* mutants. While around 80% of 3×FG mutants contained unfertilized oocytes, only around 50% of worms in the *wago-4* mutant strains showed this phenotype (Fig. 3H). This suggests that additional factors may contribute to the more severe phenotype in 3×FG mutants. Nevertheless, our results indicate that reduced WAGO-4 partitioning in the germ granules is likely a major contributor to the fertility-related phenotype observed in 3×FG mutants.

It has previously been reported that *wago-4* loss-of-function mutants exhibit the Mortal Germline (Mrt) phenotype—a gradual decline in fertility over generations at elevated temperatures, ultimately resulting in sterility (Wan et al, 2018; Seroussi et al, 2023). To test whether 3×FG mutants exhibit a similar Mrt phenotype, we cultured worms at 25 °C for multiple generations. By the eighth generation, 3×FG mutant worms became sterile (Fig. 3I). *wago-4(ko)* and mitochondria-localized WAGO-4 strains showed similar Mrt phenotypes, reaching sterility after eight and seven generations, respectively, while WT worms remained fertile throughout the experiment.

In addition, we have investigated the genetic interaction between FG mutations in GLH proteins and WAGO-4 mislocalization. We generated a double mutant combining the 3×FG background with the *mts::gfp::wago-4* mislocalization mutant. Our fecundity tests revealed no additive effect at either 20 °C or 26 °C (Fig. EV3F,G). This suggests that FG mutations in GLH proteins and WAGO-4 mislocalization likely act through the same pathway.

Together, these findings demonstrate that *wago-4* mutants and 3×FG mutants exhibit similar germline phenotypes and that the FG mutations in GLH proteins and WAGO-4 mislocalization likely impair fertility through the same pathway. We cannot entirely exclude unintended gain-of-function effects resulting from WAGO-4 mislocalization to ectopic sites (cytoplasm, mitochondria, plasma membrane), and we acknowledge that GLH-4 and potentially other granule proteins are also mislocalized in the 3×FG mutant, which could lead to a complex phenotype. However, the consistency of phenotypes across the diverse WAGO-4 mislocalization contexts and the loss-of-function mutant, and similarity with 3×FG mutant phenotypes suggest that impaired WAGO-4 granule localization is likely the primary contributor to the observed fertility defects. This indicates that the WAGO-4 function is compromised when its normal enrichment in germ granules is disrupted.

## 3×FG mutants display altered mRNA expression profiles indicative of impaired WAGO-4 function

WAGO-4 has been described in two roles: it is essential for the inheritance of sRNAs targeting genes silenced by exogenous RNAi, while its endogenous sRNAs primarily target protein-coding genes associated with germline expression (Xu et al, 2018; Wan et al, 2018; Charlesworth et al, 2021; Seroussi et al, 2023). Despite lacking the key catalytic residues (DEDH) required for Ago-mediated slicer activity (Yigit et al, 2006), WAGO-4 is essential for silencing mRNAs targeted by exogenous RNAi, as evidenced by examining the expression of these target mRNAs in WT versus *wago-4* null mutants (Xu et al, 2018). The mechanism of this silencing remains unclear, but it has been proposed that WAGO-4 represses its endogenous germline targets through a similar, yet unidentified, pathway (Weiser and Kim, 2019; Sundby et al, 2021). If this is the case and the WAGO-4 function is compromised in 3×FG mutants, we would expect changes in the mRNA expression of WAGO-4 targets, which could provide insights into the molecular basis of the 3×FG mutant phenotype.

To investigate this, we performed mRNA sequencing on 3×FG worms grown at 20 °C and 26 °C and compared their expression profiles to WT. Since the phenotype is primarily observed at 26 °C, we expected more pronounced differences at this temperature. Surprisingly, the opposite was true: more differentially expressed genes were detected at 20 °C than at 26 °C. A similar pattern was observed in the expression profiles of *wago-4(Y611E)* and *wago-4(ko)* mutants compared to WT, suggesting this result was not specific to 3×FG mutants (Fig. EV4A). Indeed, the differences in mRNA expression between animals grown at different temperatures were larger than the differences between WT and mutant animals (Fig. EV4B). We speculate that worms experience substantial stress at 26 °C, triggering a generalized stress response in both WT and mutant animals, potentially masking more subtle differences in expression between genotypes. Supporting this hypothesis, we observed greater gene expression variation between 26 °C and 20 °C within the same strain (WT: 2309 differentially regulated genes, 3×FG: 1431, *wago-4(Y611E)*: 2281, *wago-4(ko)*: 2614; data provided in Dataset EV1) than between WT and mutants at 26 °C (3×FG: 443 differentially regulated genes, *wago-4(Y611E)*:385, *wago-4(ko)*: 434; Fig. EV4A). Additionally, analysis of the genes commonly regulated at 26 °C across all strains revealed significant enrichment for stress response genes among the upregulated genes ($P = 1.39e\text{-}17$). For the downregulated genes, ribosome biogenesis and translation initiation factors were most enriched ($P = 1.39e\text{-}51$) (data provided in Dataset EV1). This pattern aligns with the reported global decrease in translation under stress to conserve energy for essential processes and stress adaptation (Holcik and Sonenberg, 2005). Conversely, while no phenotype is apparent at 20 °C, WAGO-4 localization and/or function are still affected in the described mutants at this temperature, potentially leading to molecular changes. Therefore, we focused our analysis on the data from worms grown at 20 °C.

We identified 634 upregulated and 831 downregulated genes in 3×FG mutants compared to WT (Fig. 4A). Both groups of genes were slightly enriched for genes associated with oogenic and germline-constitutive expression ($\log_2$ enrichment <1.2), while spermatogenic genes were slightly depleted among the upregulated genes ($\log_2$ enrichment $-0.37$) and strongly depleted among the

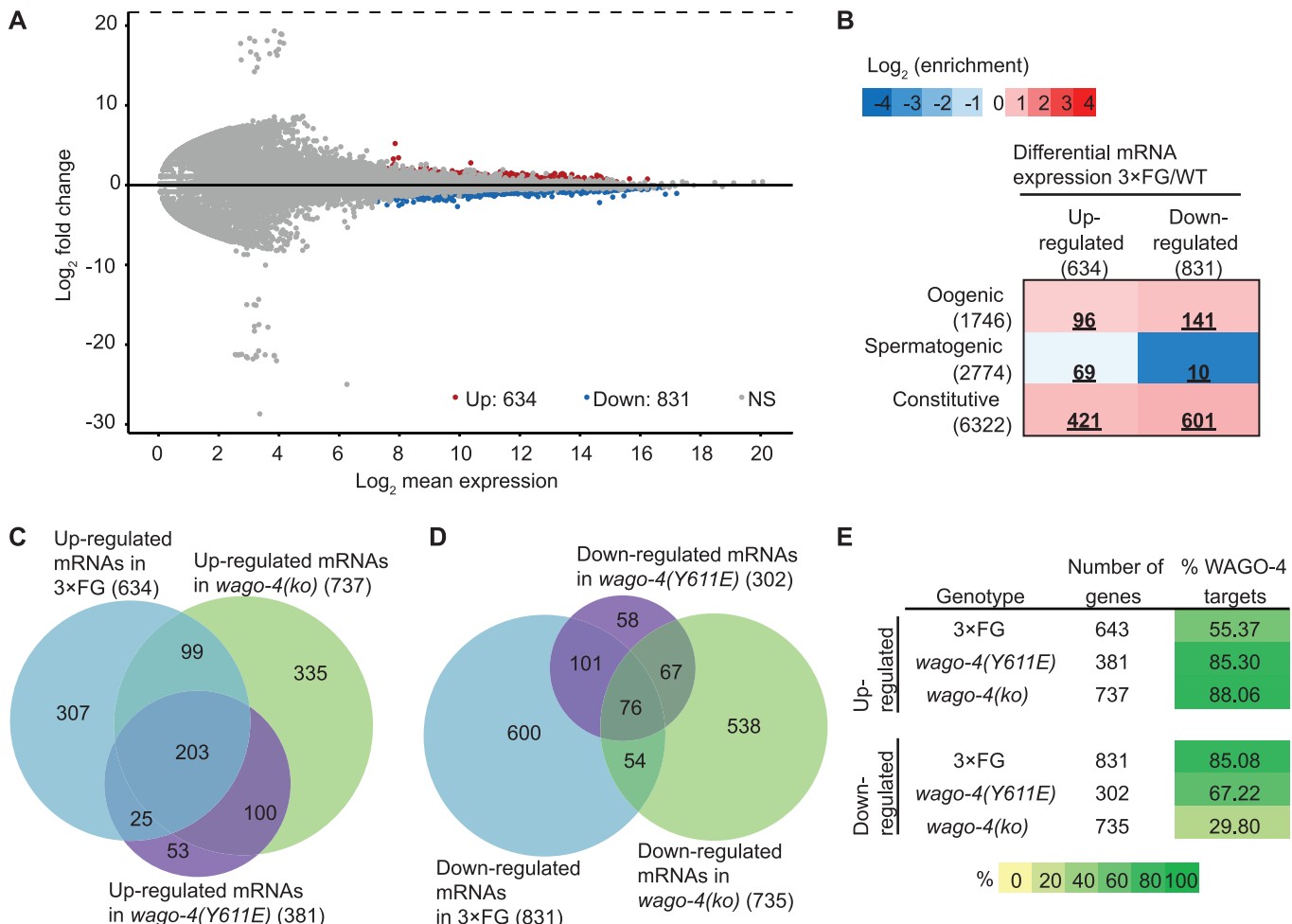

**Figure 4. 3×FG mutants display altered mRNA expression profiles indicative of impaired WAGO-4 function.**

(A) MA plot representing differential gene expression (3×FG vs.WT) based on mRNA sequencing from worms grown at 20 °C. Red dots indicate upregulated genes, blue dots indicate downregulated genes, and gray dots represent genes without significant changes in expression. The number of regulated genes is indicated at the bottom ($P < 0.05$). (B) Enrichment analysis of genes classified as oogenic, spermatogenic, or germline-constitutive (Ortiz et al, 2014) compared to genes up- or downregulated in 3×FG vs. WT based on mRNA sequencing. Log2 enrichment is indicated by color according to the scale shown, and significant enrichment or depletion (p < 0.05, Fisher's exact test) is highlighted in bold and underlined. (C) Venn diagram showing the overlap of upregulated genes ($P < 0.05$) from mRNA sequencing for three genotypes: 3×FG, wago-4(Y611E), and wago-4(ko). Numbers in brackets indicate the size of each gene group. (D) Venn diagram showing the overlap of downregulated genes ($P < 0.05$) from mRNA sequencing for three genotypes: 3×FG, wago-4(Y611E), and wago-4(ko). Numbers in brackets indicate the size of each gene group. (E) Table presenting the percentage of up- or downregulated genes in 3×FG, wago-4(Y611E), and wago-4(ko) that are identified as WAGO-4 targets according to Seroussi et al (2023). Percentages are highlighted with yellow and green hues. The total number of up- and downregulated genes is indicated for each condition.

downregulated genes (log2 enrichment −3.53) (Fig. 4B; germline expression patterns based on Ortiz et al, 2014).

Given our hypothesis that WAGO-4 function might be compromised in 3×FG mutants, we compared differentially expressed genes in 3×FG with those in wago-4(Y611E) and wago-4(ko) strains. Upregulated genes in wago-4(Y611E) and wago-4(ko) extensively overlapped, and approximately half of the upregulated genes in 3×FG overlapped with those in the wago-4 mutants (Fig. 4C). The overlap between downregulated genes across the three strains was smaller, consistent with the current understanding that WAGO-4 primarily acts as a silencing factor (Fig. 4D). In WAGO-4 loss-of-function conditions, we expect upregulation of its direct targets, whereas downregulation likely reflects secondary effects. Supporting this, a high percentage of upregulated genes in wago-4(Y611E) and wago-4(ko) mutants were previously reported as WAGO-4 targets (Seroussi et al, 2023), while

this percentage was lower for downregulated genes (Fig. 4E). Interestingly, in 3×FG mutants, the percentage of WAGO-4 targets was higher among downregulated genes than among upregulated genes. These findings indicate that WAGO-4 function is likely compromised in 3×FG mutants, though this condition does not represent a loss of WAGO-4 function.

When we closely examined the downregulated genes in 3×FG mutants, we identified 22 histone genes with reduced expression, spanning various histone types without clear specificity (Fig. EV4C). Histone downregulation has been reported in several RNAi pathway mutants, including csr-1 and prg-1, and linked to chromatin instability and fertility defects (Avgousti et al, 2012; Barucci et al, 2020; Cecere, 2021). Consistent with this, our mass spectrometry analysis of 3×FG mutants at 26 °C showed reduced levels of multiple histones and CDL-1, a factor required for histone

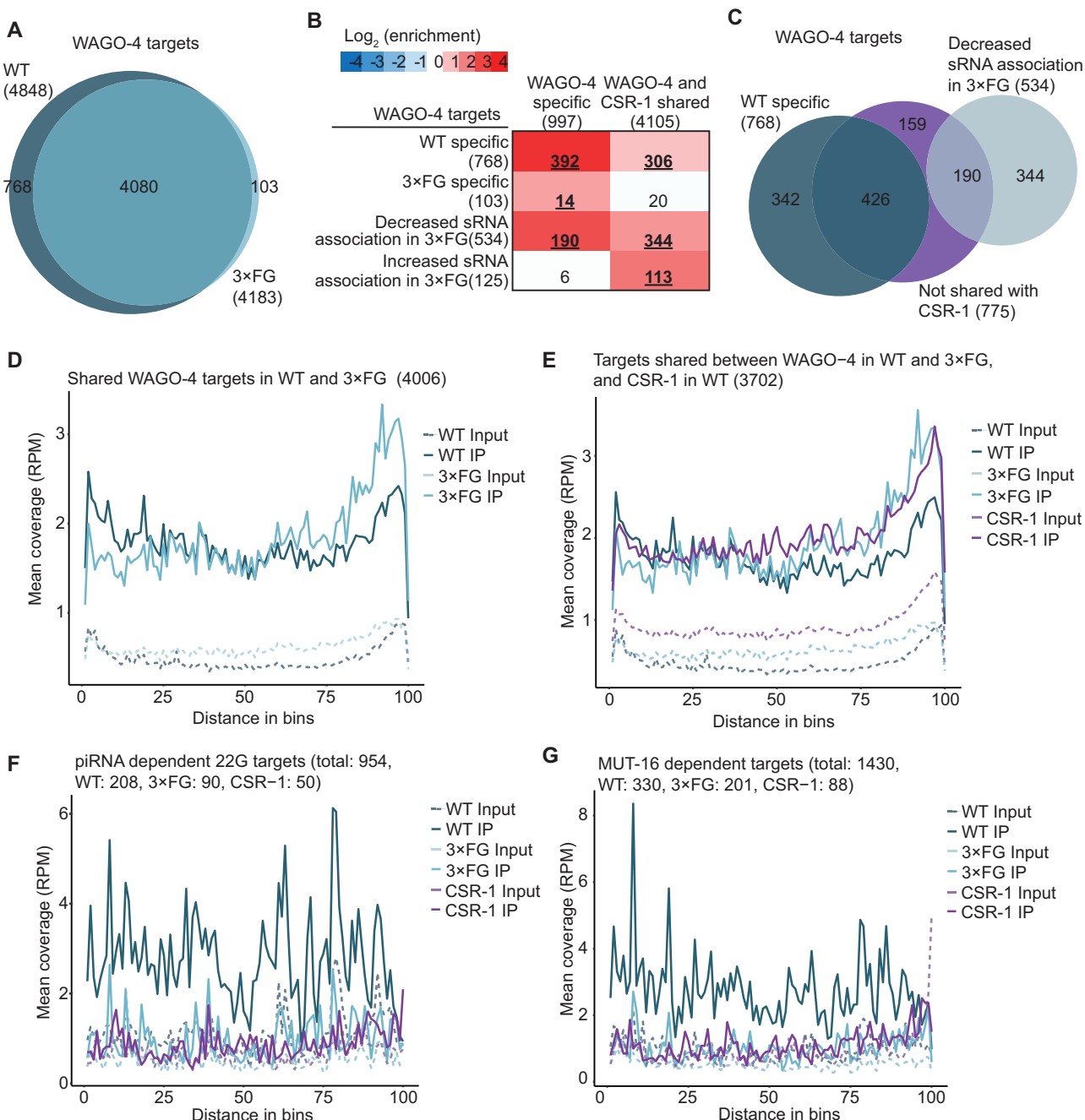

mRNA translation (Fig. EV4D). These results suggest a potential contribution of histone downregulation to the fertility phenotype.

Our findings indicate that 3×FG mutants have a distinct mRNA expression profile likely influenced by compromised WAGO-4 function. This altered expression profile may contribute to the reduced fertility observed in 3×FG mutants at elevated temperatures through multiple mechanisms, potentially involving significant downregulation of histone genes, which have been previously linked to impaired fertility.

## The 3×FG mutation selectively disrupts WAGO-4's association with sRNAs complementary to non-CSR-1-targets

To investigate how the WAGO-4 function is affected in the 3×FG mutant, we assessed the repertoire of sRNAs associated with WAGO-4 and compared it to that of WT. We performed sRNA sequencing coupled with immunoprecipitation (IP) of WAGO-4 (Fig. EV5A) from samples of worms grown at 20 °C. Our data

**Figure 5. The 3×FG mutation selectively disrupts WAGO-4's association with sRNAs complementary to non-CSR-1-targets.**

(A) Venn diagram showing the overlap of WAGO-4 targets identified in WT and 3×FG worms based on sRNA sequencing after immunoprecipitation (IP). Targets were defined as having at least 5 RPM and an IP/input fold change ≥2 in each of three biological replicates. Results represent worms grown at 20 °C. Numbers in brackets indicate the size of each group. (B) Enrichment of WAGO-4 sRNA targets in four groups (WT-specific, 3×FG-specific, targets with a decrease in WAGO-4-associated sRNAs in 3×FG, and targets with an increase in WAGO-4-associated sRNAs in 3×FG) compared to WAGO-4 specific targets (i.e., not shared with CSR-1), or targets shared with CSR-1 (Seroussi et al, 2023). Log$_2$ enrichment is indicated by color according to the scale shown, and significant enrichment or depletion (p < 0.05, Fisher's exact test) is highlighted in bold and underlined. (C) Venn diagram showing the overlap of WAGO-4 targets in three groups: (1) WT-specific targets (present in WT but absent in 3×FG), (2) WAGO-4 targets not shared with CSR-1, and (3) targets with a decrease in WAGO-4-associated sRNAs in 3×FG. Numbers in brackets indicate the size of each group. (D) Metagene profiles for sRNAs complementary to protein-coding gene WAGO-4 targets shared between WT and 3×FG. Targets were normalized by size, partitioned into 100 bins, and the mean coverage (RPM) for each bin was plotted. Solid lines represent IP values, and dashed lines represent input values. The number of targets is indicated in brackets. (E–G) Metagene profiles for sRNAs complementary to protein-coding gene WAGO-4 and CSR-1 targets. CSR-1 targets were identified from CSR-1 IP in WT worms (Seroussi et al, 2023). Size-normalized targets were partitioned into 100 bins, and mean coverage (RPM) for each bin was plotted. Solid lines represent IP values, and dashed lines represent input values. The number of targets analyzed for each genotype is indicated in brackets: (E) genes identified as WAGO-4 (both in WT and 3×FG) and CSR-1 targets; (F) genes identified as piRNA-dependent 22 G targets (Barucci et al, 2020) and (G) MUT-16-dependent targets (Chen et al, 2024). Source data are available online for this figure.

showed that 96% of WAGO-4 targets identified in WT were also previously detected as WAGO-4 targets (Seroussi et al, 2023). In 3×FG mutants, WAGO-4 bound a largely overlapping set of targets with WT, sharing 4,080 targets (Fig. 5A). However, the number of targets identified was higher in WT (4,848) than in 3×FG (4,183). Among these, 768 targets were identified only in WT (~16%; referred to as WT-specific targets), and 103 targets were identified only in 3×FG (~2%; referred to as 3×FG-specific targets). In addition, 534 targets showed significant decreases in WAGO-4-associated 22G-RNAs, while 125 targets showed increases (Fig. EV5B).

We analyzed whether WAGO-4 targets with differential 22G-RNA targeting in 3×FG mutants exhibited differential gene expression. We calculated the enrichment of WT-specific targets, 3×FG-specific targets, and targets with down- or upregulated sRNAs relative to up- and downregulated genes (Fig. EV5C). Most groups showed enrichment, except for targets with increased WAGO-4-associated sRNAs, which had no significant association with upregulated genes in 3×FG. Interestingly, the highest enrichment (log$_2$ enrichment 2.48) was observed between targets with increased WAGO-4-associated sRNAs and downregulated genes in 3×FG, a group that includes histone genes (Fig. EV4C). However, none of these groups accounted for more than 1% of either up- or downregulated genes.

To further understand how the WAGO-4 function is altered in 3×FG mutants, we examined whether WT-specific targets have distinctive characteristics. WAGO-4 and CSR-1 are known to share a large set of targets (Charlesworth et al, 2021; Seroussi et al, 2023), and 84% of WAGO-4 targets in our dataset overlapped with previously reported CSR-1 targets (Seroussi et al, 2023) (Fig. EV5D). We analyzed whether WT-specific or 3×FG-specific targets were enriched for WAGO-4-specific targets (i.e., targets that are not shared with CSR-1) or WAGO-4/CSR-1 shared targets (Fig. 5B). To avoid bias caused by differences in target identification between WT and 3×FG, we defined WAGO-4-specific and WAGO-4/CSR-1 shared targets based on published data (Seroussi et al, 2023). Interestingly, WT-specific targets were highly enriched for WAGO-4-specific targets (log$_2$ enrichment 3.35), accounting for 392 of 768 WT-specific targets. In addition, WAGO-4-specific targets were strongly enriched for targets with decreased WAGO-4-associated 22G-RNAs in 3×FG mutants (log$_2$ enrichment 2.83) (Fig. 5B). Together, nearly 80% of WAGO-4-specific targets identified in WT (based on Seroussi et al, 2023) were either WT-

specific or targets with decreased 22G-RNAs in 3×FG (Fig. 5C). These findings suggest that in 3×FG mutants, WAGO-4 binds 22G-RNAs antisense to targets not shared with CSR-1 less efficiently. Consistent with this, almost all targets with increased WAGO-4-associated sRNAs in 3×FG mutants (113/125) were shared between WAGO-4 and CSR-1 (Fig. 5B).

We next compared WAGO-4 22G-RNA targeting profiles between WT and 3×FG. Metagene analysis revealed that WAGO-4 in 3×FG showed slightly reduced targeting at the 5' end and increased targeting at the 3' end of mRNAs compared to WT (Figs. 5D and EV5E). Since WAGO-4 in 3×FG mutants seemed to bind 22G-RNAs of CSR-1 shared targets more efficiently than WAGO-4-specific targets, we compared WAGO-4 22G-RNA targeting profiles in WT and 3×FG to the CSR-1 22G-RNA targeting profile. For this analysis, we used published sRNA sequencing data from CSR-1 immunoprecipitation (Seroussi et al, 2023). Notably, the WAGO-4 targeting profile in 3×FG more closely resembled the CSR-1 profile than the WT WAGO-4 profile (Figs. 5E and EV5F). These data suggest that in 3×FG mutants, WAGO-4 preferentially associates with 22G-RNAs that target transcripts co-targeted by CSR-1. This raises the possibility that WAGO-4 utilizes at least two distinct 22G-RNA biogenesis/loading pathways, one being the CSR-1-associated pathway. In 3×FG mutants, the CSR-1-associated pathway may be more extensively used than in WT.

Germ granules are proposed to serve as hubs for sRNA biogenesis and loading of germline Argonautes (Sundby et al, 2021; Phillips and Updike, 2022). However, CSR-1 22G-RNA biogenesis and loading were shown to occur in the cytoplasm, independent of germ granule formation (Singh et al, 2021). The precise location of WAGO-4 22G-RNA biogenesis and loading remains unclear. Given that WAGO-4 partitioning to the germ granules is strongly reduced in 3×FG mutants, with WAGO-4 primarily localized in the cytoplasm, it is possible that the cytoplasmic CSR-1-associated 22G-RNA biogenesis and loading pathway remains accessible to WAGO-4. In this scenario, any WAGO-4 sRNA loading dependent on its localization to germ granules would likely become less efficient.

To explore this possibility, we investigated whether WAGO-4 binds 22G-RNAs antisense to targets shown to be germ granule-dependent. Specifically, we examined two groups: (1) piRNA-dependent 22G-RNA targets (Barucci et al, 2020) shown to require P granule formation (Singh et al, 2021) and (2) MUT-16-dependent

targets, which require the formation of Mutator foci (Chen et al, 2024) (Fig. 5F,G). In WT, approximately 22% of the targets from both groups were identified as WAGO-4 targets (208/954 for group 1 and 330/1430 for group 2). In 3×FG mutants, WAGO-4 targeting was reduced for both groups, with fewer targets identified (90 vs. 208 for group 1, and 201 vs. 330 for group 2) and lower mean coverage for the identified targets compared to WT. Furthermore, the WAGO-4 targeting profiles in 3×FG more closely matched the CSR-1 profiles than the WAGO-4 profiles in WT for both target groups.

Together, these findings suggest that WAGO-4 could use multiple 22G-RNA biogenesis and loading pathways. In 3×FG mutants, where it is largely cytoplasmic, WAGO-4 may increasingly rely on the cytoplasmic pathway shared with CSR-1, which leads to the alternation of its overall targeting profile and changes in gene regulation.

## Discussion

This study reveals that the Ago WAGO-4 requires germ granule localization for accurate sRNA loading and gene regulation. To assess its functional dependence on germ granules, we leveraged a unique system—3×FG mutants, in which FG repeats in Vasa-like GLH proteins were perturbed, and WAGO-4 was mislocalized from germ granules to the cytoplasm. This system minimized confounding factors, as granule integrity was largely preserved, WAGO-4 itself was not mutated, and FG mutations did not impair GLH function. Previous studies have shown that WAGO-4's endogenous targets are primarily protein-coding mRNAs, with significant overlap with CSR-1 targets (Xu et al, 2018; Wan et al, 2018; Charlesworth et al, 2021; Seroussi et al, 2023). In 3×FG mutants with reduced WAGO-4 partitioning into germ granules, WAGO-4 exhibited selective reduction in loading of 22G-RNAs antisense to its unique targets, while those antisense to CSR-1-shared targets remained largely unaffected. Germ granules are proposed hubs for sRNA biogenesis and Argonaute loading (Sundby et al, 2021; Phillips and Updike, 2022); however, while CSR-1 loads 22G-RNAs in the cytoplasm (Singh et al, 2021), the precise site of WAGO-4 loading remains unclear. Our findings support a model in which WAGO-4 uses at least two distinct sRNA loading pathways: a cytoplasmic pathway shared with CSR-1 and a germ granule-dependent pathway required for targeting unique transcripts. When WAGO-4 mislocalizes from germ granules, it likely relies more on the cytoplasmic pathway, altering its sRNA repertoire and impairing its function.

GLH proteins are Vasa-like DEAD-box RNA helicases essential for RNA remodeling and sRNA pathway regulation (Gruidl et al, 1996; Kuznick et al, 2000; Marnik et al, 2019; Chen et al, 2020, 2022; Dai et al, 2022). While their helicase activity is critical for sRNA pathways, FG repeats play a distinct role. FG repeats have been proposed to interact with nuclear pore complex (NPC) FG domains to anchor P granules (Updike et al, 2011), but their role in recruiting GLH proteins or other components to perinuclear granules remains unclear (Marnik et al, 2019; Chen et al, 2020). Our 3×FG mutants allowed us to dissect FG-repeat function (Ribbeck and Görlich, 2002; Patel et al, 2007): we found that P granules were formed and remained perinuclear in these mutants, suggesting GLH-FG repeats are not strictly required for granule

formation or anchoring. Consistently, a recent study shows that efficient reduction of perinuclear P granules requires the removal of an FG-containing nucleoporin and the P granule factor MIP-1 (Thomas et al, 2025). Our data demonstrate that FG repeats in GLH proteins are crucial for the proper localization of WAGO-4 to germ granules, although the underlying mechanism remains unclear. They may directly recruit WAGO-4, similar to how Drosophila GW182 recruits Ago1 to P bodies (Eulalio et al, 2009), or alternatively, fine-tune condensate composition by modulating hydrophobicity. Supporting the latter, previous work indicated that P granule integrity relies on hydrophobic interactions (Updike et al, 2011). Our in vitro reconstitution experiments suggest that FG repeats contribute to WAGO-4 recruitment into phase-separated granules, but their loss alone is insufficient to fully abrogate WAGO-4 recruitment to granules. While further research is needed to elucidate the precise mechanism of FG-repeat function in germ granules, current data support the idea that they play a subtle role in modulating granule hydrophobicity and, consequently, protein composition. This raises the possibility that Vasa in *C. elegans* has two distinct functions: one as an RNA helicase and another in maintaining condensate composition through FG-repeat-mediated interactions. Our bioinformatic analysis revealed FG/GF dipeptide enrichment in the IDR regions of Vasa orthologs across diverse animal lineages, including Chordata (Fig. EV6). Given that Vasa is a conserved germ granule component (Voronina et al, 2011), there is an intriguing possibility that FG repeats in Vasa may have an evolutionarily conserved role in condensate organization and selective protein recruitment.

WAGO-4 has two key functions: inheriting sRNAs that maintain silencing of exogenous RNAi targets and regulating germline-expressed protein-coding genes via endogenous 22G-RNAs (Xu et al, 2018; Wan et al, 2018; Charlesworth et al, 2021; Seroussi et al, 2023). In this study, we investigated the functional significance of WAGO-4 germ granule localization. Spatial separation of sRNA biogenesis, loading, and/or targeting has been proposed as a likely mechanism for regulating Agos with overlapping targets, such as CSR-1 and WAGO-4 (Sundby et al, 2021). Since CSR-1 is not implicated in exogenous RNAi inheritance, we focused on WAGO-4's endogenous function. Despite lacking predicted slicer-catalytic residues (Yigit et al, 2006), WAGO-4 is essential for silencing exogenous RNAi targets, as evidenced by their reduced expression (Xu et al, 2018). Consistently, *wago-4* null and sRNA-binding-defective mutants show upregulation of endogenous target mRNAs, supporting its role in gene silencing of its endogenous targets. This suggests that WAGO-4 promotes mRNA decay, though additional regulatory mechanisms, such as translational repression, cannot be excluded. Interestingly, WAGO-4 mislocalization in 3×FG mutants led to pronounced target downregulation, suggesting a functional shift rather than a simple loss of activity. We propose that this shift results from a disrupted balance between WAGO-4 and CSR-1 in germline gene regulation, as discussed below.

WAGO-4 and CSR-1 share ~4000 mRNA targets, are co-expressed, and enriched in germ granules, but their interplay remains unclear. CSR-1 has been suggested to license gene expression, and WAGO-4 to act as a silencer, but how this antagonism is regulated remains unknown (Wedeles et al, 2013; Seth et al, 2013; Xu et al, 2018; Sundby et al, 2021). When we examined the sRNA targeting profiles of WAGO-4 and CSR-1 for

their shared targets, we observed subtle differences, as CSR-1-associated sRNAs are more enriched at 3' ends of target mRNA compared to WAGO-4-associated sRNAs. In 3×FG mutants, WAGO-4 shifts toward the CSR-1 sRNA repertoire: (1) binding of sRNAs antisense to WAGO-4 targets not shared with CSR-1 decreases, and (2) even for shared targets, WAGO-4 binds more sRNAs toward the 3' end, mirroring the CSR-1 targeting pattern rather than that of WT WAGO-4. This suggests that the spatial positioning of WAGO-4 and CSR-1 is important for their correct sRNA loading. A recent study in *C. elegans* demonstrated that HRDE-2 is required for Ago HRDE-1 localization to germ granules (specifically SIMR foci) and correct sRNA loading, supporting a role for granule enrichment in proper Argonaute-sRNA loading (Chen and Phillips, 2024). While other factors, such as sRNA modifications like untemplated 3' end uridylation, may influence Ago affinity (van Wolfswinkel et al, 2009; Xu et al, 2018; Charlesworth et al, 2021), our findings emphasize that subcellular localization plays a key role in defining WAGO-4 vs. CSR-1 sRNA specificity.

In 3×FG mutants, WAGO-4's shift toward the CSR-1 sRNA repertoire is accompanied by a functional shift: WAGO-4 targets were predominantly downregulated rather than upregulated. Notably, 94% of these downregulated targets are also CSR-1 targets, suggesting that this shift may interfere with CSR-1's expected licensing function. We speculate that WAGO-4 binding CSR-1-associated sRNAs could lead to excessive repression of genes that CSR-1 would normally license for expression, disrupting the equilibrium between WAGO-4 and CSR-1 pathways. The fact that CSR-1 mutants are sterile indicates that precise regulation of its targets is crucial for fertility (Yigit et al, 2006; Claycomb et al, 2009). However, the molecular link between this dysregulation and the 3×FG mutant phenotype is not completely clear. Our data suggest that stress responses may obscure mutant-specific transcriptional changes at high temperatures, where the phenotype manifests. We speculate that transcriptional shifts observed at lower temperatures might be compensated under permissive conditions but become detrimental at elevated temperatures. For example, among downregulated genes in 3×FG mutants are multiple histone genes, and histones are normally positively regulated by CSR-1 (Avgousti et al, 2012; Charlesworth et al, 2021). As histone depletion has been associated with impaired fertility (Avgousti et al, 2012; Charlesworth et al, 2021; Barucci et al, 2020), and we detected reduced histone protein levels at high temperatures, this may be a contributing factor for the observed fertility defects.

Our findings indicate the importance of balancing WAGO-4 and CSR-1 activities in germline gene regulation, but the precise mechanisms underlying their interplay—whether dictated by intrinsic sRNA properties, target binding site locations, intracellular activity sites, or other factors—remain to be elucidated. In addition, WAGO-4 localization may play a role in exogenous RNAi inheritance, as CSR-1 has not been implicated in this pathway. Although not explored in this study, granule localization could be crucial for WAGO-4 loading sRNAs involved in exogenous RNAi. This line of investigation, though beyond the scope of the current work, represents an important direction for future research.

Together, our findings highlight the role of spatial compartmentalization in sRNA pathways and Ago function. Our data indicate that two co-expressed Agos with overlapping targets differentially rely on spatially distinct sRNA biogenesis and loading

pathways, maintaining their antagonistic roles in gene regulation. In addition, weak hydrophobic interactions appear to fine-tune membrane-less compartment composition. These results emphasize the need to precisely dissect the role of protein partitioning within biomolecular condensates. Advancements in understanding the mechanisms governing precise protein intra-condensate partitioning could enable targeted control of protein localization and selective function modulation, with potential applications in disease modeling and therapeutic development.

# Methods

**Reagents and tools table**

| Reagent/resource | Reference or source | Identifier or catalog number |
| --- | --- | --- |
| **Experimental models** | | |
| *C elegans* strains | Table EV2 | |
| Insect cell culture Gibco Sf9 | Thermo Fisher Scientific | 11496015 |
| **Recombinant DNA** | | |
| Plasmid pAD895 | Holzer et al, 2022 | |
| Plasmid pMS050 (mScarlet sequence) | Addgene | #91826 |
| Plasmid pIM001 (mEGFP sequence) | This study | |
| **Antibodies** | | |
| **Oligonucleotides and other sequence-based reagents** | | |
| gRNAs | IDT | Table EV3 |
| **Chemicals, enzymes, and other reagents** | | |
| FM4-64 dye | Invitrogen | T13320 |
| Pierce Streptavidin magnetic beads | Thermo Fisher Scientific | 88816 |
| Streptavidin-Alexa Fluor 488 | Thermo Fisher Scientific | S32354 |
| TRI Reagent | Sigma | T9424-100ML |
| SuperaseIN | Invitrogen | AM2694 |
| ChromoTek GFP-Trap Magnetic Agarose | ChromoTek | gtma |
| Anti-FLAG M2 Magnetic Beads | Merck | M8823 |
| poly-A+ mRNA | Takara | 636101 |
| cOmplete EDTA-free Protease Inhibitor Cocktail | Roche | 04693132001 |
| Ni-NTA Agarose | Protino | 745400 |
| Q Sepharose resin | Cytiva | GE17-0510-01 |
| HiLoad 16/60 Superdex 200 column | Cytiva | 28989335 |
| **Software** | | |
| Fiji | Schindelin et al 2012 | http://fiji.sc/Fiji |
| IMARIS | Oxford Instruments | |
| RStudio (version 2024.12.0.467) | Posit Software | |
| GraphPad Prism (version 10.2.3) | GraphPad Software, Inc. | |

| Reagent/resource | Reference or source | Identifier or catalog number |
|---|---|---|
| **Other** | | |
| iST kit | PreOmics | P.O.00027 |
| EnzChek Phosphate Assay Kit | Invitrogen | E6646 |
| RNeasy kit | QUIAGEN | 74134 |
| Ultra II Directional RNA Library Prep | New England Biolabs | E7765 |
| NEBNext Small RNA Library Prep Set for Illumina | New England Biolabs | E7330L |
| Illumina NovaSeqX | Illumina | |
| Illumina Miniseq | Illumina | |
| Illumina Novaseq 6000 | Illumina | |
| Olympus IX83 inverted microscope fitted with a spinning-disk scan head (Yokogawa CSU-W1) | Olympus/Yokogawa | |

## Methods and protocols

### C. elegans strains and maintenance

All *C. elegans* strains were maintained on 9 cm plates containing Nematode Growth Media with OP50 bacterial lawn using standard methods (Brenner, 1974) at 20 °C. The strains were either generated using CRISPR/Cas-based technology, by crossing or obtained from the CGC. The list of the strains used in this study is provided in Table EV2, and the list of gRNAs in Table EV3. For CRISPR gene targeting, crRNAs, tracrRNA, Cas9, Cas12a, and HDR donor oligos were obtained from IDT. Larger insertions containing introns were amplified from plasmids (mEGFP, mScarlet-I, and TurboID–information in the Reagents and Tools table) or from the *C. elegans* genomic DNA (MTS and PH– sequences provided in Table EV4) using PCR with primers that incorporated 35–50 nucleotide homology arms. Donor constructs for *glh-1* and *glh-4* containing F → A mutations (including 145 bp or 420 bp homology arms) were synthesized by Genewiz and amplified by PCR. All strains were verified by sequencing.

### Fecundity test

We assessed the fecundity of the worms at 20 °C and 26 °C. For both assays, the animals were maintained at 20 °C. In the 20 °C assay, 20 L4 worms from each strain were singled. Each worm was transferred to a fresh plate every 24 h for five days, with the plates kept at 20 °C. On the fourth day after each transfer, the daily number of progeny from each worm was recorded. For the 26 °C assay, 10 L4 worms were initially picked from the population maintained at 20 °C and placed on a single plate at 26 °C. When their progeny reached the L4 stage, 20 L4 worms from each strain were singled and kept at 26 °C throughout the experiment. The daily progeny count for each worm was recorded on the third day after transfer. For both experiments, the total progeny count for each worm was calculated by summing the daily counts. Worms that did not survive until the end of the experiment were excluded from the analysis. To ensure consistency across conditions, all the strains for a single experiment were thawed and assayed at the same time.

### Mating assay

To determine if the fecundity defects in worms arise from defective sperm, we conducted a mating assay. The worms were maintained at 20 °C. For mating, males of the SS6 strain (PGL-3::mCherry) were used, enriched by maintaining a 2:6 ratio of hermaphrodites to males. Assayed hermaphrodites were prepared by placing 10 L4 worms from each of the tested strains (N2, 2×FG, and 3×FG) on a single plate at 26 °C until their progeny reached the L4 stage. The assay was performed by singling 10 L4 worms that developed at 26 °C and adding four SS6 strain males to each plate. All plates were then maintained at 26 °C throughout the experiment. All five worms (one hermaphrodite and four males) were transferred to a fresh plate every 24 h. Progeny scoring was performed as in the fecundity test at 26 °C. Progeny positive for the PGL-3::mCherry marker were detected in all the plates which were not empty, indicating successful mating.

### Mortal germline assay

The Mortal germline assay was performed as described (Seroussi et al, 2023). Worms were initially maintained at 20 °C. Five L4 worms of each strain were placed on each plate and incubated at 25 °C. Five plates for each strain were used to establish separate lines. Every 3 days, five L4 worms from a new generation were transferred to a fresh plate. This was repeated for each line until there were no progeny left to pick five L4 worms from. All the assayed lines reached the mortal stage within eight generations except for the N2 control line.

### Unfertilized oocyte phenotype scoring

To quantify the occurrence of the unfertilized oocyte phenotype, whole-mount DAPI staining was performed on worms grown at 20 °C and 26 °C. For the 20 °C assay, 9 L4 worms from each strain were placed on a fresh plate. The P0 adults were removed after 24 h, and the worms were washed off the plates after four days at 20 °C. For the 26 °C assay, 10 L4 worms from populations maintained at 20 °C were placed on a fresh plate and kept at 26 °C. The P0 adults were removed after 24 h, and the worms were grown for three days at 26 °C before being washed off the plates. The worms were collected in low-binding 1.5-mL tubes using M9 buffer and washed three times with PBS (137 mM NaCl, 2.7 mM KCl, 10 mM $Na_2HPO_4$, and 1.8 mM $KH_2PO_4$) to remove bacteria. Fixation was performed in 100% ethanol for 5 min at room temperature. The fixed worms were washed twice with PBS and stained with DAPI (4 μg/mL) for 5 min at room temperature. Following three PBS washes, the fixed worms were mounted in an antifade mounting medium (Vectashield, Vector Laboratories) on slides.

To score the phenotype, the uteri of the worms were examined, and each worm was categorized into one of three groups: (1) only embryos present, (2) only unfertilized oocytes present, and (3) a combination of embryos and unfertilized oocytes present. Within category 3, the severity of the phenotype varied, ranging from cases with multiple embryos and one unfertilized oocyte to cases with multiple unfertilized oocytes and one embryo. To avoid bias, worms were scored in one of two ways: (1) if there were ~50 worms on a slide, all the worms on the slide were scored, or (2) if there were significantly more than 50 worms on the slide, an overview image of the sample was taken, and 50 worms were chosen in advance at random for scoring. Three or four experiments, each with ~50 worms, were conducted to calculate the average

occurrence of unfertilized oocytes. To ensure the same conditions, all the strains were thawed and assayed at the same time. Microscopy was performed using the spinning disc fluorescent microscope (described in detail in the Live Imaging section) with a ×40 dry objective.

### Live imaging

The images were captured using an Olympus IX83 inverted microscope fitted with a spinning-disk scan head (Yokogawa CSU-W1) operating at 4000 rpm. Imaging was facilitated by either a Teledyne Photometrics Prime 95B sCMOS camera with a pixel size of 11 μm or a Hamamatsu Orca Fusion CMOS camera with a pixel size of 6.5 μm. We used 100×/1.4 oil-immersion objective (Olympus UPLSAPO) or a 40×/0.75 air objective (Olympus UPLSAPO). Illumination was provided by Coherent OBIS lasers at wavelengths of 405 nm, 488 nm, 561 nm, and 640 nm.

The imaged worms were always at the young adult stage. For 20 °C experiments, L4 worms were picked from the plate, with the worms continuously maintained at 20 °C, approximately 16 h before imaging. For 26 °C experiments, parent worms (maintained at 20 °C) were picked at the L4 stage and placed at 26 °C. Imaging was performed when the oldest progeny had just reached the young adult stage (after 3 days). Parent worms were removed from the plates after 24 h to ensure they were excluded from the analysis. For imaging, microscopy slides were prepared by placing up to 10 worms on a single agarose pad in a drop of 2% levamisole. To minimize imaging artifacts, worms were imaged within 30 min. Imaging parameters were adjusted individually for each fluorescently labeled protein and temperature, based on signal strength. Control and mutant worms expressing the same fluorescently labeled protein were always imaged during the same imaging session and under identical imaging conditions (laser power, exposure settings, and z-slice thickness).

For colocalization analysis (Fig. 3A), worms were anesthetized using 0.15% sodium azide to achieve complete immobilization, which is critical for accurate analysis. When sodium azide was used, worms were never exposed on the slide for more than 15 min to minimize potential effects on experimental outcomes.

To determine if the eggshell is formed in the large-cell structures deposited in the worms' uteri, the following protocol was followed: the worms were grown for one generation at 26 °C, as for the fecundity test. Worms were dissected by opening the uterus and embryos and/or large-cell structures were released in the media which supports survival of early embryos (60% Leibowitz L-15 media, 25 mM HEPES pH 7.4, 0.5% Inulin, 20% heat-inactivated fetal bovine serum) supplemented with polystyrene beads (25 μm in diameter), FM4-64 fluorescent dye (Invitrogen, final concentration 5 μM) and DAPI (1 μg/mL). Live embryos were imaged using the microscope system described above. To ensure the conditions supported early embryo development, several WT 1 cell stage embryos were monitored to develop until the 4-cell stage. The large-cell structures did not divide during this time.

### Image analysis

Image analysis was conducted using the Fiji image-processing package (http://fiji.sc/Fiji). Displayed images are represented with the same contrast settings for each WT and mutant strain, which express the same fluorescently labeled protein and were developed at the same temperature.

IMARIS software was used for the 3D analysis of images obtained by live imaging with the microscope system described above (Fig. EV2A–C). For the calculation of P granule numbers and total volume (Fig. EV2A,B), only those germ cell nuclei for which the imaged Z-stack covered the full volume were used. Seven different nuclei from three different animals were scored per strain ($n = 21$). IMARIS was used to calculate Pearson's coefficient for colocalized volume between PGL-3::mCherry and GFP::ZNFX-1 in WT and 3×FG background (Fig. EV2C). The coefficient was calculated for a single worm containing ~30 pachytene nuclei per image, and there were six different worms analyzed per condition.

To quantify fluorescent intensities in perinuclear regions (Fig. EV2F,G), we used a Fiji-macro including deep learning for the nuclei detection. Quantification was performed on a single confocal slice representing the middle plane of the analyzed nucleus. A custom-trained model was used in Cellpose to mark the outlines of the nuclei. From these segmentations, we created ring-shaped selections covering the perinuclear region, as indicated in Fig. EV2F. Intensities were then measured along the selections, and the SD of the signal was calculated.

To quantify WAGO-4 enrichment in P and Z granules (Fig. 3B), we analyzed fluorescence micrographs from live worms expressing GFP::WAGO-4 together with PGL-3::mCherry or RFP::ZNFX-1 in WT and 3×FG backgrounds, grown at 20 °C. To ensure optimal image quality, several preparatory steps were performed. First, regions of interest (ROIs) were cropped to maximize the number of Pachytene germline nuclei, minimizing signal interference from surrounding structures. The analysis included approximately 35 nuclei for P granules ($35.3 \pm 2.8$) and 30 nuclei for Z granules ($29.7 \pm 2.7$). To correct for minor sample movements during acquisition, image stacks were realigned using Napari-Pystackreg. Signal enhancement and noise reduction were performed using Huygens deconvolution software. ROIs for P and Z granules were defined using Fiji macros, segmenting PGL-3::mCherry and RFP::ZNFX-1 signals, respectively. A Laplacian of Gaussian (LoG) filter was applied to normalize intensity differences and improve thresholding. Granule volumes were well captured using 54 confocal planes with 0.1 μm steps. For accurate intensity ratio calculations, GFP::WAGO-4 mean intensity was measured in P or Z granules as well as in the cytoplasm, and the enrichment ratio was determined for each Z-stack slice. This analysis was conducted for each condition using images from three individual worms.

### Proximity labeling

For the 20 °C experiment, worms continuously maintained at 20 °C were synchronized by bleaching adults only (separated from laid embryos by sedimentation) and plated on 15 cm NGM plates with 4-ml of OP50 bacteria. They were grown until they reached the young adult stage. For the 26 °C experiment, parent worms (maintained at 20 °C) were synchronized by chunking embryos onto 9-cm NGM plates seeded with OP50 bacteria. The plates were incubated at 20 °C until the worms reached the L4 stage, then transferred to 26 °C. After 24 h, the adults were bleached. Plates were kept at 26 °C until the worms reached the young adult stage. Four to six 15 cm plates were used for each replicate in all experiments, and three biological replicates were prepared for each strain. Worms were washed three times with M9 buffer and once with Milli-Q water. The pellet was resuspended in RIPA buffer (Sigma) and flash-frozen by dropping it into liquid nitrogen. The frozen samples were stored at –80 °C until ready for streptavidin pulldown.

The frozen samples were ground in a mortar with liquid nitrogen. Protein lysates were prepared by thawing the frozen powder, sonicating (low setting for 3 min, 15 s ON/15 s OFF), and centrifuging for 15 min at 20,000×g. All three steps were performed at 4 °C. Lysates were incubated with streptavidin magnetic beads (Pierce, Thermo Scientific) at 4 °C overnight. The beads were washed nine times, each wash lasting 5 min with rotation. The first wash was performed at 4 °C, and the remaining washes at room temperature. The wash buffers used were as follows: RIPA buffer (Sigma) for the first wash; a buffer containing 50 mM Tris-HCl pH 7.5, 150 mM NaCl, 1 mM EDTA, and 2% SDS for the second wash; a buffer containing 50 mM Tris, pH 7.4, 500 mM KCl, 1 mM EDTA, and 1% NP-40 for the third wash; a buffer containing 0.1 M $Na_2CO_3$, pH 11.5, and 0.1% NP-40 for the fourth wash; a buffer containing 2 M Urea, 10 mM Tris, pH 8 for the fifth wash; and a buffer containing 20 mM Tris, pH 7.5, and 137 mM NaCl for the sixth through ninth washes. The beads were transferred to a fresh tube between the sixth and seventh washes to reduce detergent contamination. After the final wash, 15% of the beads were kept for quality check using western blot, and the rest were dry-frozen for subsequent mass spectrometry (MS) analysis.

Frozen beads were resuspended in 50ul of 100 mM ammonium bicarbonate (ABC), supplemented with 400 ng of lysyl endopeptidase (Lys-C, Fujifilm Wako Pure Chemical Corporation) and incubated for 4 h on a Thermo-shaker with 1200 rpm at 37 °C. The supernatant was transferred to a fresh tube and reduced with 0.5 mM Tris 2-carboxyethyl phosphine hydrochloride (TCEP, Sigma) for 30 min at 60 °C and alkylated in 3 mM methyl methanethiosulfonate (MMTS, Fluka) for 30 min at room temperature, protected from light. Subsequently, the sample was digested with 400 ng trypsin (Trypsin Gold, Promega) at 37 °C overnight. The digest was acidified by the addition of trifluoroacetic acid (TFA, Pierce) to 1%. A similar aliquot of each sample (20%) was analyzed by LC-MS/MS.

### Whole lysate preparation for MS analysis

The worm samples were prepared the same way as for the proximity labeling experiment and frozen in liquid nitrogen. The frozen samples were ground in a mortar with liquid nitrogen. Protein lysates were prepared by thawing the frozen powder, sonicating (low setting for 3 min, 15 s ON/15 s OFF), and centrifuging for 15 min at 20,000×g. Protein concentration was estimated using the Bradford assay (Bio-Rad). 100 μg of the protein per sample was used to proceed with preparation for the MS analysis using the iST kit (PreOmics), according to the manufacturer's protocol.

### MS protocol and analysis

The nano HPLC system (UltiMate 3000 RSLC nano system) was coupled to an Orbitrap Exploris 480 mass spectrometer, equipped with a FAIMS pro interface and a Nanospray Flex ion source (all parts Thermo Fisher Scientific).

Peptides were loaded onto a trap column (PepMap Acclaim C18, 5 mm×300 μm ID, 5 μm particles, 100 Å pore size, Thermo Fisher Scientific) at a flow rate of 25 μl/min using 0.1% TFA as mobile phase. After loading, the trap column was switched in line with the analytical column (PepMap Acclaim C18, 500 mm×75 μm ID, 2 μm, 100 Å, Thermo Fisher Scientific). Peptides were eluted using a flow rate of 230 nl/min, starting with the mobile phases 98% A (0.1% formic acid in water) and 2% B (80% acetonitrile, 0.1%

formic acid) and linearly increasing to 35% B over the next 120 or 180 min. This was followed by a steep gradient to 95% B in 5 min, stayed there for 5 min, and ramped down in 2 min to the starting conditions of 98% A and 2% B for equilibration at 30 °C.

The Orbitrap Exploris 480 mass spectrometer was operated in data-dependent mode, performing a full scan (m/z range 350–1200, resolution 60,000, normalized AGC target 100%) at three different compensation voltages (CV −45 V, −60 V, and −75 V), followed by MS/MS scans of the most abundant. MS/MS spectra were acquired using an isolation width of 1.0 m/z or 1.2 m/z, normalized AGC target 200% or 100%, minimum intensity set to 25,000, HCD collision energy of 30, maximum injection time of 100 ms or 30 ms, and resolution of 30,000 or 15,000. Precursor ions selected for fragmentation (include charge state 2-6) were excluded for 45 s. The monoisotopic precursor selection (MIPS) mode was set to peptide, and the exclude isotopes feature was enabled.

The Q Exactive HF-X mass spectrometer was operated in data-dependent mode, using a full scan (m/z range 380–1500, nominal resolution of 60,000, target value 1E6) followed by MS/MS scans of the 10 most abundant ions. MS/MS spectra were acquired using normalized collision energy of 28, isolation width of 1.0 m/z, resolution of 30.000, target value of 1E5, maximum fill time 105 ms. Precursor ions selected for fragmentation (include charge states 2-6) were put on a dynamic exclusion list for 60 s. Additionally, the minimum AGC target was set to 5E3, and the intensity threshold was calculated to be 4.8E4. The peptide match feature was set to preferred, and the exclude isotopes feature was enabled.

For peptide identification, the RAW files were loaded into Proteome Discoverer (version 2.5.0.400, Thermo Scientific). All MS/MS spectra were searched using MSAmanda v2.0.0.19924 (Dorfer et al, 2014). The peptide mass tolerance was set to ±10 ppm and fragment mass tolerance to ±10 ppm, the maximum number of missed cleavages was set to 2, using tryptic enzymatic specificity without proline restriction. Peptide and protein identification were performed in two steps. For an initial search, the RAW-files were searched against the Uniprot reference database for *C. elegans* subspecies (19,834 sequences; 8,141,223 residues), supplemented with common contaminants and sequences of tagged proteins of interest using beta-methylthiolation or Iodoacetamide derivative, respectively, on cysteine as a fixed modification. The result was filtered to 1% FDR on the protein level using the Percolator algorithm (Käll et al, 2007) integrated in Proteome Discoverer. A sub-database of proteins identified in this search was generated for further processing. For the second search, the RAW-files were searched against the created sub-database using the same settings as above and considering the following additional variable modifications: oxidation on methionine, deamidation on asparagine and glutamine, phosphorylation on serine, threonine and tyrosine, glutamine to pyro-glutamate conversion at peptide N-terminal glutamine, and acetylation on protein N-terminus. The localization of the post-translational modification sites within the peptides was performed with the tool ptmRS, based on the tool phosphoRS (Taus et al, 2011). Identifications were filtered again to 1% FDR on protein and PSM level, additionally, an Amanda score cut-off of at least 150 was applied. Proteins were filtered to be identified by a minimum of 2 PSMs in at least 1 sample. Protein areas have been quantified using IMP-apQuant (Doblmann et al, 2019) by summing unique and razor peptides and applying intensity-based absolute quantification (Schwanhäusser et al, 2011) with subsequent

normalization based on the MaxLFQ algorithm (Cox et al, 2014). Quantifications of FG mutant proteins were manually recomputed by summing up all peptides which are the same for both WT and mutant protein, in order to represent them as a single protein in volcano plots, in contrast to reporting two entries (WT & mutant) with razor proteins assigned to a single form and quantification of the other form by the single unique peptide at best. Peptides shared between GLH homologs were excluded from this procedure. The resulting quantifications were sum-normalized per sample, and statistical significance of differentially abundant proteins was determined using limma (Smyth, 2004).

### PRM MS analysis

PRM MS analysis was used to determine the expression level of WAGO-4 in the WAGO-4 KO mutant. The processing of the samples before the run on the mass spectrometer was the same as for the whole lysate analysis. The UltiMate 3000 RSLC coupled to the Orbitrap Eclipse mass spectrometer, equipped with the NanoFlex Ion source and the FAIMS Pro Duo (Thermo) were used. Peptides were loaded onto a trap column (PepMap C18, 5 mm×300 µm ID, 5µm particles, 100 Å pore size, Thermo) by using 0.1% TFA at a flow rate of 25 µl/min. The trap column was switched in line with the analytical column (Aurora Ultimate 25 cm×75 µm C18 UHPLC column, Ionopticks). The analytical column was heated up to 50 °C in the Sonation column oven (PRSO-V2). Electrospray voltage was set to 2.4 kV. Peptides were eluted at 300 nl/min using a binary 60-min gradient within the complete 105 min LC-MS/MS run, including column equilibration.

The gradient starts with the mobile phases: 98% A (water/formic acid, 99.9/0.1, v/v) and 2% B (water/acetonitrile/formic acid, 19.92/80/0.08, v/v/v), increases to 35% B over the next 90 min, followed by a gradient in 5 min to 90% B. A column equilibration was done with 2% B for 25 min.

The Orbitrap Eclipse mass spectrometer was operated by a mixed MS method which consisted of one full scan ($m/z$ range 350–1200; resolution 15,000; target value 100%; FAIMS CV value −45 V) followed by the PRM of time scheduled targeted peptides from an inclusion list (isolation window 0.8 m/z; normalized collision energy (NCE) 32; resolution 30,000, AGC target 200%). Each precursor had a specific FAIMS CV value (−45 V or −60 V or −75 V) and the maximum injection time set up.

A scheduled PRM method (sPRM) development, data processing, and manual evaluation of results were performed in Skyline (v. 22.2.0.351) (MacLean et al, 2010). Spectra of unique peptides of the proteins of interest (WAGO-4) and 5 normalization controls (EF1A, SAHH, RL22, RL27, RL36) were recorded. Each unique peptide used for relative quantification of the protein of interest had to have at least three fragment ions.

For each protein of interest and normalization control, the measured peptide areas were summed up. The obtained value for each protein of interest was divided by the value corresponding to the individual normalization control.

### Immunofluorescence

Immunofluorescence microscopy was used to detect biotinylation in strains expressing TurboID-tagged to PGL-3. Microscope slides were coated with 0.1% poly-lysine (Sigma). Worms were dissected to extrude gonads and then fixed with 1% formaldehyde for 5 min at room temperature. A freeze-crack method was performed by placing the slides in liquid nitrogen. Membrane permeabilization was achieved by incubating the slides in 100% methanol at −20 °C for 10 min. After three washes with PBS-T (137 mM NaCl, 2.7 mM KCl, 10 mM Na$_2$HPO$_4$, and 1.8 mM KH$_2$PO$_4$, 0.1% Tween-20), the samples were blocked with 1% BSA at room temperature for 30 min. Streptavidin-Alexa Fluor 488 (Invitrogen) was incubated at 4 °C overnight at a dilution of 1:1000 in 1% BSA. The samples were then washed three times in PBS-T and stained with DAPI (10 µg/mL) for 10 min at room temperature. After an additional three washes in PBS-T, the samples were mounted with an antifade mounting medium (Vectashield, Vector Laboratories).

### Recombinant protein expression and purification

Recombinant GLH-1, GLH-1 F → A, GLH-4, GLH-4 F → A constructs were produced in Sf9 insect cells (Commercial cell line: Gibco Sf9, Thermo Fisher Scientific) using a baculovirus infection system. The insect cells were collected approximately 72 h post-infection and subsequently lysed. The constructs were purified from the lysates through a combination of Ni-NTA affinity, anion-exchange, and size-exclusion chromatography, as described below.

Insect cells expressing GLH-1, GLH-1 F → A, GLH-4, or GLH-4 F → A tagged with 6xHis-mEGFP were resuspended in a lysis buffer containing 25 mM HEPES pH 7.5, 300 mM KCl, 10 mM imidazole, and 1 mM DTT, supplemented with a Complete EDTA-free protease inhibitor cocktail (Roche). The cells were lysed using a sonicator, and the lysates were centrifuged at 21,000×g using a JS-13.1 rotor (Beckman-Coulter). Ni-NTA Agarose resin (Macherey-Nagel) was then added to the supernatant to capture the 6xHis-tagged proteins. The resin was washed with 25 mM HEPES pH 7.5, 300 mM KCl, and 20 mM imidazole, and the proteins were eluted with 25 mM HEPES pH 7.5, 300 mM KCl, and 250 mM imidazole.

For anion-exchange chromatography, the eluate was diluted 1:6 in 25 mM Tris pH 8.0 and 1 mM DTT, and 4 ml of Q Sepharose resin (GE Healthcare) was added to bind the protein of interest. After ~30 min of incubation, the resin was washed with 25 mM Tris pH 8.0 and 1 mM DTT buffers containing increasing concentrations of KCl (200 mM, 250 mM, 300 mM, 350 mM, 400 mM, 500 mM, and 1 M). The fractions containing 250–400 mM KCl were pooled. The pooled fractions underwent further purification via size-exclusion chromatography using a HiLoad 16/60 Superdex 200 column (GE Healthcare) in 25 mM HEPES, pH 7.5, 300 mM KCl, and 1 mM DTT.

To obtain untagged GLH-1, GLH-1 F → A, GLH-4, or GLH-4 F → A, the 6xHis and mEGFP tags were cleaved from the C-terminus of the recombinant proteins using 6xHis-tagged TEV protease. The mixture was then incubated with Ni-NTA Agarose resin to remove the 6xHis-tagged mEGFP and TEV protease. Proteins were then concentrated and kept at −70 °C. Protein preparations were checked by SDS-PAGE and MS.

PGL-1, PGL-3, and PGL-3::mEGFP were expressed and purified as described (Jelenic et al, 2024).

### mScarlet::WAGO-4 purification from worms

Worms expressing 3xFLAG::mScarlet::WAGO-4 (modification introduced in the native *wago-4* locus) were maintained at 20 °C, synchronized by bleaching adults only (excluding laid embryos), and plated on 15-cm NGM plates with 4 ml of OP50 bacteria. They were grown until they reached the young adult stage and then washed off the plates with M9 buffer. Worms were washed three times with M9 buffer and once with Milli-Q water. The worm pellet

was resuspended at a 1:1.5 (v/v) ratio in EDTA buffer, which contained 10% glycerol, 10 mM EDTA, 30 mM HEPES, 100 mM potassium acetate, 2 mM DTT, and 0.1% NP-40, supplemented with protease, RNase, and phosphatase inhibitors: 2× cOmplete EDTA-free protease inhibitor cocktail (Roche), 1:100 SuperaseIN (Thermo Fisher), and 1:100 phosphatase inhibitor cocktails 2 and 3 (Sigma). The resuspended pellet was flash-frozen by adding it drop by drop into liquid nitrogen, and the frozen samples were stored at –80 °C until lysate preparation.

The frozen samples were ground in a mortar with liquid nitrogen. Protein lysates were prepared by thawing the frozen powder, sonicating it at a low setting (3 min, 15 s ON/15 s OFF), and centrifuging it at 17,000×g for 15 min, with all steps performed at 4 °C. Once thawed, the samples were handled continuously at 4 °C or kept on ice. The lysates were then incubated with Anti-FLAG M2 magnetic beads (Merck) at 4 °C for 1.5 h with rotation to bind WAGO-4. After binding to the magnetic beads, the beads were washed three times with a buffer containing 25 mM HEPES pH 7.5, 200 mM KCl, 5% glycerol, 0.05% NP-40, and 1 mM DTT. Afterward, the protein was eluted with the buffer containing 25 mM HEPES pH 7.5, 150 mM KCl, 1% glycerol, 1 mM DTT, and 250 ng/μL of 3xFLAG peptide for 25 min, rotating at 4 °C. The eluate was concentrated in a centrifugation filter with 50 kDa cutoff, and the buffer was exchanged (25 mM HEPES pH 7.5, 150 mM KCl, 1% glycerol, 1 mM DTT). The final protein sample containing full-length protein and an excess of 3xFLAG peptide removed was confirmed by SDS-PAGE followed by Coomassie InstantBlue (Abcam) protein staining.

### In vitro reconstitution experiment

To reconstitute minimal P granules, we used recombinantly purified proteins (PGL-1, PGL-3, GLH-1, and GLH-4) and commercial pure poly-A+ mRNA (Takara). Proteins and mRNA were mixed before phase separation was triggered by diluting the salt (KCl) to a final concentration of 100 mM in 25 mM HEPES pH 7.5, 1 mM DTT. Proteins were combined in stoichiometric ratios as described (Saha et al, 2016) with the following final concentrations after triggering phase separation: 5 μM PGL-3 (95% untagged and 5% PGL-3::mEGFP), 6.7 μM PGL-1, 1 μM GLH-1, and 1 μM GLH-4. Two types of condensates were formed: one containing WT GLH-1 and GLH-4 and another containing F → A mutated versions of these proteins (same mutations as in C. elegans strains, Fig. 1A). We then added equal amounts of mScarlet::WAGO-4 purified from worms, or the buffer as a negative control (same buffer as the mScarlet::WAGO-4 storage buffer) to both condensates. Condensates were imaged on the pretreated slides (Alberti et al, 2018) using the above-described imaging system. Images were analyzed in Fiji using the following approach: images were first corrected for uneven illumination and camera noise. For each image, six regions of interest (ROIs; 2.47×2.47 μm) were selected within distinct condensates, and six additional ROIs of the same size were placed outside the condensates. ROIs were selected based solely on the green channel (PGL-3::GFP) to visualize condensed and dilute regions while avoiding bias. Signal intensity in the red channel (mScarlet::WAGO-4 or buffer in negative controls) was then measured for all ROIs. To calculate enrichment, the intensity of each inside-condensate ROI was divided by the average red-channel intensity of the six outside ROIs (dilute phase). Since ROIs were predetermined based on the green channel, any

ROI that contained a mScarlet::WAGO-4 aggregate was excluded from analysis at this stage. This analysis was performed on six independent images for each condition.

### In vitro ATPase assay

To test the enzymatic activity of GLH-1 (WT and F to A mutants), we used the EnzChek Phosphate Assay Kit (Invitrogen) according to the manufacturer's protocol. The protein concentration used for the assay was 20 μM. The enzymatic activity for each protein was assessed both in the presence and absence of total yeast RNA (Sigma), with each condition tested in triplicate. The activity was determined by measuring the absorbance at 360 nm every two minutes with the BioTek Synergy H1 plate reader. The relative absorbance was calculated by subtracting the absorbance of the no-enzyme control from that of the enzyme reaction.

### mRNA sequencing

Worms were synchronized and washed following the Proximity Labeling protocol for both 20 °C and 26 °C conditions. After three washes with M9 buffer, the worm pellet was mixed with 1 mL of TRI Reagent (Sigma) and kept at –80 °C. Before RNA isolation, samples were subjected to three freeze–thaw cycles using dry ice and vortexing in between. RNA was isolated following the TRI Reagent manufacturer's protocol: phase separation with BCP, RNA precipitation with isopropanol, wash with 75% ethanol, and resuspension in Nuclease-free water. RNA samples were treated with DNase I (NEB) using the manufacturer's protocol and finally cleaned up using the RNeasy kit (QIAGEN).

RNA sequencing was performed in the VBCF (Vienna BioCenter Core Facilities) Next Generation Sequencing (NGS) Facility (www.viennabiocenter.org/facilities). Library preparation, quality control, and sequencing were performed by the Facility. Total RNA input samples were subjected to automated ribonucleic depletion using rRNA depletion kit Ribovanish prior to library preparation using a starting amount of 500 ng total RNA. The library was prepared using Ultra II Directional RNA Library Prep Kit (NEB, Cat #E7765) for all samples. The fragment size of the libraries was assessed using the NGS HS analysis kit and a fragment analyzer system (Agilent). The library concentrations were quantified with a KAPA Kit (Roche).

The libraries were sequenced on 1 lane of an Illumina NovaSeqX 10B XP flowcell, PairedEnd 150. RNA sequencing data have been analyzed using the RNA-Seq pipeline of the IMP/IMBA Bioinfo core facility (ii-bioinfo@imp.ac.at). The pipeline is based on the nf-core/rnaseq pipeline [https://doi.org/10.5281/zenodo.1400710] and is built with NextFlow (Ewels et al, 2020).

### sRNA sequencing

Sample preparation and sRNA sequencing were performed as described (Renaud et al, 2023), and downstream analyses were conducted as described (Seroussi et al, 2023). To determine differential expression of small RNAs in mutant strains compared to wild-type, DESeq2 was used (version 3.14; Love et al, 2014). The volcano plots were generated with EnhancedVolcano (Blighe et al, 2018).

### Bioinformatic analysis of Vasa orthologs

Vasa and DDX4 homologs were collected in UniProt reference proteomes, ENSEMBL and NCBI protein databases with NCBI blast+ (UniProt Consortium, 2021; NCBI Resource Coordinators,

2018; Yates et al, 2020; Altschul et al, 1997) A maximum likelihood phylogenetic tree was calculated with IQ-TREE 2 (v.2.2.0, Minh et al, 2020), using an alignment of selected family members with mafft (v7.505, -linsi method, Katoh and Toh, 2008) and extracting the conserved region covering Drosophila melanogaster vasa from residues 199 to 624 (UniProt accession: P09052) with Jalview (Waterhouse et al, 2009) The phylogenetic tree was infered with standard model selection using ModelFinder (Kalyaanamoorthy et al, 2017) and ultrafast bootstrap (UFBoot2) support values (Hoang et al, 2018). The tree was visualized in iTOL (v7, Letunic and Bork, 2021). To determine "FG/GF"-rich regions, we first identified G-rich segments in protein sequences using fLPS (Harrison, 2017). These fragments were used to calculate dipeptide frequencies with POPPs (Wise, 2002), with a background distribution from the eukaryotic UniProt reference proteomes and a probability threshold of 0.005.

### Statistical analysis

Statistical analysis of large datasets (including MS, mRNA sequencing, sRNA sequencing, and bioinformatic analysis of Vasa orthologs) was performed as described in the respective sections. Gene set enrichment analysis was conducted using Fisher's exact test, implemented in RStudio (version 2024.12.0.467). To account for multiple comparisons, the Benjamini-Hochberg false discovery rate (FDR) correction was applied to the P values. To assess the enrichment of Regulated Gene Sets in temperature-related differential gene expression data, the online tool WormCat was used (Holdorf et al, 2020). All remaining statistical analyses were performed in GraphPad Prism (version 10.2.3). Statistical tests were selected based on experimental design, number of groups, sample size, data distribution, and the appropriate unit of replication.

For comparisons involving more than two groups, we used either the Kruskal–Wallis test or one-way ANOVA. In fecundity assays, data were frequently non-normally distributed and were analyzed using the Kruskal–Wallis test followed by Dunn's post hoc correction. Group differences were visualized using letter annotations in figures with multiple comparisons for visual clarity; all corresponding P values are provided in Dataset EV2. In one experiment (Fig. 3G), 5/21 pairwise comparisons yielded uncorrected P values below 0.05, but became non-significant after Dunn's correction due to >20× P value inflation. In this case, the Benjamini–Krieger–Yekutieli (BKY) false discovery rate (FDR) correction was additionally applied (indicated in the figure), and all uncorrected, Dunn-adjusted, and FDR-adjusted values are reported in Dataset EV2. For phenotypic scoring (e.g., unfertilized oocyte quantification), the percentage of animals in each category was calculated per experiment, with each replicate comprising ~50 animals. Variability between replicates was low and results were consistent. Group means from three independent experiments were analyzed using one-way ANOVA. Statistical significance was not indicated in the graphs for clarity, but corresponding P values are provided in Dataset EV2.

For pairwise comparisons, the Mann–Whitney test was used unless otherwise noted. This non-parametric test was selected due to frequent deviations from normality (assessed using D'Agostino–Pearson, Anderson–Darling, Shapiro–Wilk and Kolmogorov–Smirnov tests), or the inability to reliably assess normality in small sample sizes. One exception was PRM-MS quantification (Fig. EV3C), where Welch's t test was applied to log-transformed intensity ratios, following standard practice in proteomics to stabilize variance and allow fold-change interpretation (Huang et al, 2020).

## Data availability

*C. elegans* strains generated in this study are available upon request, and some are also accessible through the CGC. Datasets and microscopy images produced in this study are available in the following external repositories: mRNA-seq data: GEO GSE293366 (https://www.ncbi.nlm.nih.gov/geo/query/acc.cgi?acc=GSE293366). sRNA-seq data: GEO GSE294568 (https://www.ncbi.nlm.nih.gov/geo/query/acc.cgi?acc=GSE294568). Mass spectrometry data: PRIDE PXD062557 (https://www.ebi.ac.uk/pride/archive?keyword=PXD062557&sortDirection=DESC&page=0&pageSize=20). Imaging dataset: BioImage Archive S-BIAD2202.

The source data of this paper are collected in the following database record: biostudies:S-SCDT-10_1038-S44318-025-00606-x.

## Peer review information

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

## Acknowledgements

We thank Shambaditya Saha for securing funding and for his support and advice throughout the project. We thank Alexander Schleiffer for his computational analysis of Vasa orthologs, and Julius Brennecke, Elly Tanaka, and Eva Schmid for their mentorship and continued support. We also thank Balashankar Pillai, Volker Nitschko, and Anoop Kumar Yadav for their support throughout the project. We are grateful to the IMP/IMBA/GMI Protein

Chemistry and BioOptics Facilities for their outstanding support. mRNA and sRNA sequencing were performed at the VBCF NGS facility and The Center for Applied Genomics at the Hospital for Sick Children. We thank Alexander Dammermann, Verena Jantsch, and Julia Garcia Baucells for sharing protocols and reagents. We are grateful to Tim Schedl, Rene Ketting and Jan Schreier for valuable discussions. We acknowledge the Caenorhabditis Genetics Center (CGC), which is funded by the NIH Office of Research Infrastructure Programs (P40 OD010440), for providing *C. elegans* strains. The work was supported by institutional funds from IMBA of the Austrian Academy of Sciences, the FWF Grant (P34278), the Natural Sciences and Engineering Research Council of Canada (RGPIN-2020-06235), and the Canadian Institutes of Health Research (PJT-178076 and PJT-186154). MSR is supported by an Ontario Graduate Scholarship.

## Author contributions

**Stela Jelenic**: Conceptualization; Resources; Formal analysis; Validation; Investigation; Visualization; Writing—original draft; Writing—review and editing. **Mathias S Renaud**: Data curation; Formal analysis; Validation; Visualization. **Samantha Del Borrello**: Investigation. **Joseph Gokcezade**: Resources. **Janos Bindics**: Resources. **Lisa Frasz**: Resources. **Philipp Czermak**: Resources. **Peter Duchek**: Resources. **Julie M Claycomb**: Conceptualization; Resources; Supervision; Funding acquisition; Validation; Writing—original draft; Writing—review and editing.

Source data underlying figure panels in this paper may have individual authorship assigned. Where available, figure panel/source data authorship is listed in the following database record: biostudies:S-SCDT-10_1038-S44318-025-00606-x.

## Disclosure and competing interests statement

The authors declare no competing interests.

# Expanded View Figures

**Figure EV1.  Effect of FG mutations on GLH protein function and reproductive outcomes.**

(**A**) Table representing the number of FG dipeptides in GLH proteins: total number of FG dipeptides outside of helicase domains and outside of helicase and zinc finger (ZF) regions are shown. All FG dipeptides outside of helicase and ZF regions (second column) were mutated. (**B**) Graph representing the number of progeny for worms of the indicated genotypes grown at 20 °C. Each dot represents the progeny count of a single worm ($n = 17$–20). Horizontal lines represent mean values, and error bars indicate SD. Statistical analysis: Kruskal–Wallis test; the corresponding *P* values are reported in Dataset EV2. Groups sharing at least one letter are statistically indistinguishable ($P < 0.05$). (**C**) Graph representing the number of progeny for worms of the indicated genotypes grown at 26 °C for one generation. This is the second replicate of the experiment shown in Fig. 1B, which exhibited a more severe phenotype. Each dot represents the progeny count of a single worm ($n = 16$–18). Horizontal lines represent mean values, and error bars indicate SD. Statistical analysis: the Kruskal–Wallis test; the corresponding *P* values are reported in Dataset EV2. Groups sharing at least one letter are statistically indistinguishable ($P < 0.05$). (**D**) Graph representing the number of progeny for worms of the indicated genotypes grown at 25 °C. Each dot represents the progeny count of a single worm ($n = 18$–20). Horizontal lines represent mean values, and error bars indicate SD. Statistical analysis: Kruskal–Wallis test; the corresponding *P* values are reported in Dataset EV2. Groups sharing at least one letter are statistically indistinguishable ($P < 0.05$). (**E**) Graph representing the number of progeny for worms of the indicated genotypes (single *glh* mutants: *glh-1(F → A)*, *glh-2(ΔFG)* and *glh-4(F → A)*) grown at 26 °C. Each dot represents the progeny count of a single worm ($n = 18$–19). Horizontal lines represent mean values, and error bars indicate SD. Statistical analysis: Kruskal–Wallis test; the corresponding *P* values are reported in Dataset EV2. Groups sharing at least one letter are statistically indistinguishable ($P < 0.05$). (**F**) Schematic representation of wild-type (WT) and mutated form DDX-19. Exon 4 which harbors most FG repeats of DDX-19 was deleted in the native locus. 4×FG mutant contains this mutation together with FG mutations in GLH-1, GLH-2, and GLH-4 proteins, as described in Fig. 1A. (**G**) Graph representing the number of progeny for worms of the indicated genotypes grown at 26 °C for one generation. Each dot represents the progeny count of a single worm ($n = 17$–18). Horizontal lines represent mean values, and error bars indicate SD. Statistical analysis: the Kruskal–Wallis test; the corresponding *P* values are reported in Dataset EV2. Groups sharing at least one letter are statistically indistinguishable ($P < 0.05$). (**H**) Graph showing relative absorbance of reactions containing GLH-1 or GLH-1(F → A) with or without RNA, normalized to a blank reaction. Reactions were prepared using the EnzCheck Phosphate Assay Kit (Thermo Fisher). Average normalized absorbance over time ($n = 3$) indicating the release of inorganic phosphate is plotted. Error bars indicate SD. (**I**) Micrographs of early embryos from WT worms (upper) and an unfertilized oocyte from 3×FG mutant worms (lower) grown at 26 °C for one generation. Samples were stained with FM-64 (plasma membrane) and DAPI (DNA). Properly formed eggshells in WT embryos (black arrow) prevent staining by FM-64 and DAPI, whereas unfertilized oocytes lack a complete eggshell, so plasma membrane (white arrows) and DNA are stained. Scale bar: 20 μm. (**J**) Quantification of unfertilized oocytes in worms grown at 20 °C. Each worm was categorized into one of three groups: (1) containing exclusively embryos, (2) containing at least one embryo and at least one unfertilized oocyte, or (3) containing exclusively unfertilized oocytes (group 3 was not observed). Data represent averages from three independent experiments ($n = 50$ worms/experiment), with error bars indicating SD. (**K**) Quantification of unfertilized oocytes in uteri of worms grown at 26 °C for one generation. Each worm was categorized into one of three groups: (1) containing exclusively embryos, (2) containing at least one embryo and at least one unfertilized oocyte, or (3) containing exclusively unfertilized oocytes. Data represent the average proportion of worms in each category from three independent experiments ($n = 48.3 ± 3.1$ worms per experiment), with error bars indicating SD. (**L**) Graph representing the number of progeny for worms of the indicated genotypes grown at 26 °C for one generation and mated with WT males expressing PGL-3::mCherry. Each dot represents the progeny count of a single worm ($n = 8$–10). Horizontal lines represent mean values, and error bars indicate SD. Statistical analysis: the Kruskal–Wallis test; the corresponding *P* values are reported in Dataset EV2. Groups sharing at least one letter are statistically indistinguishable ($P < 0.05$).

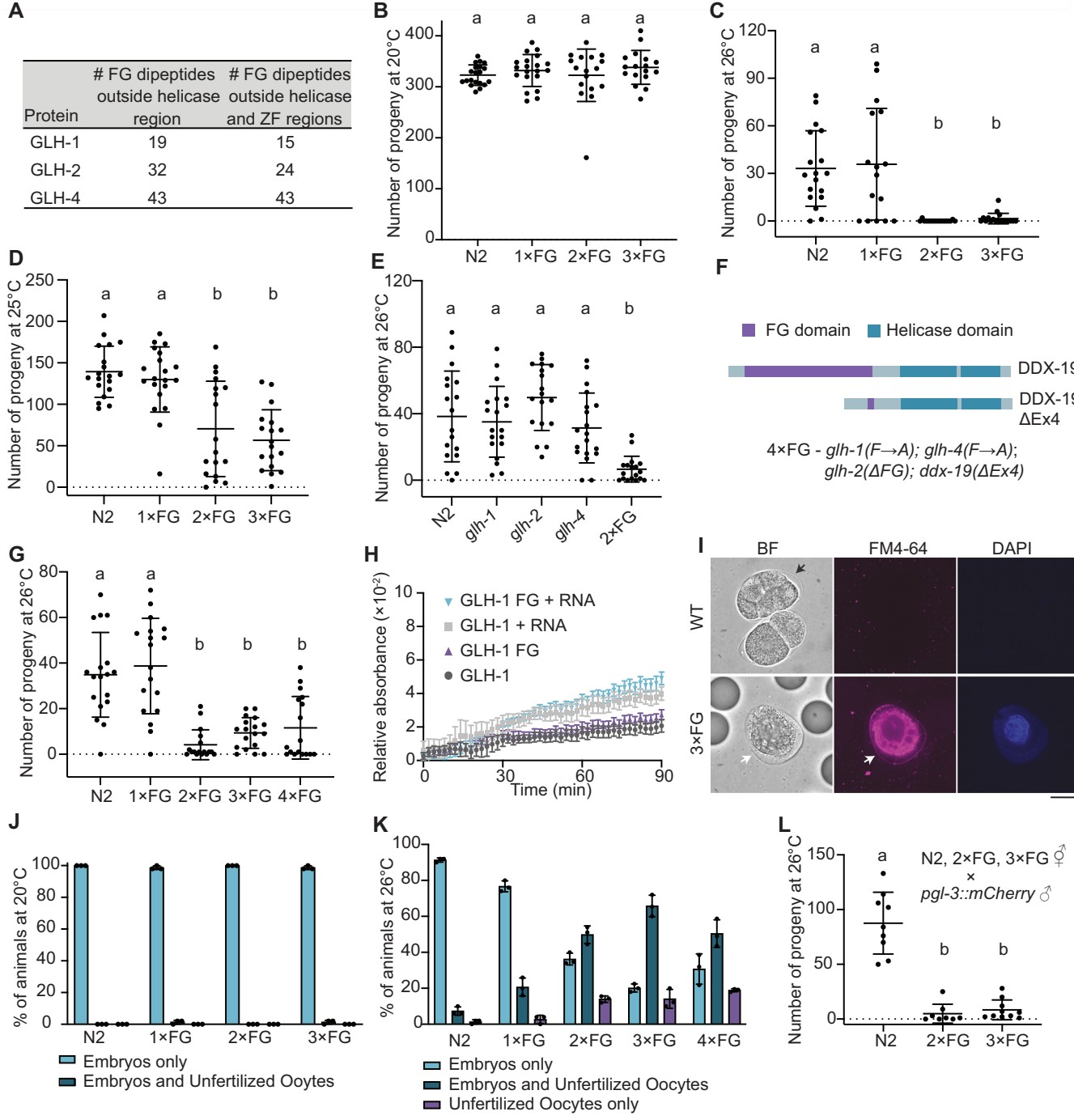

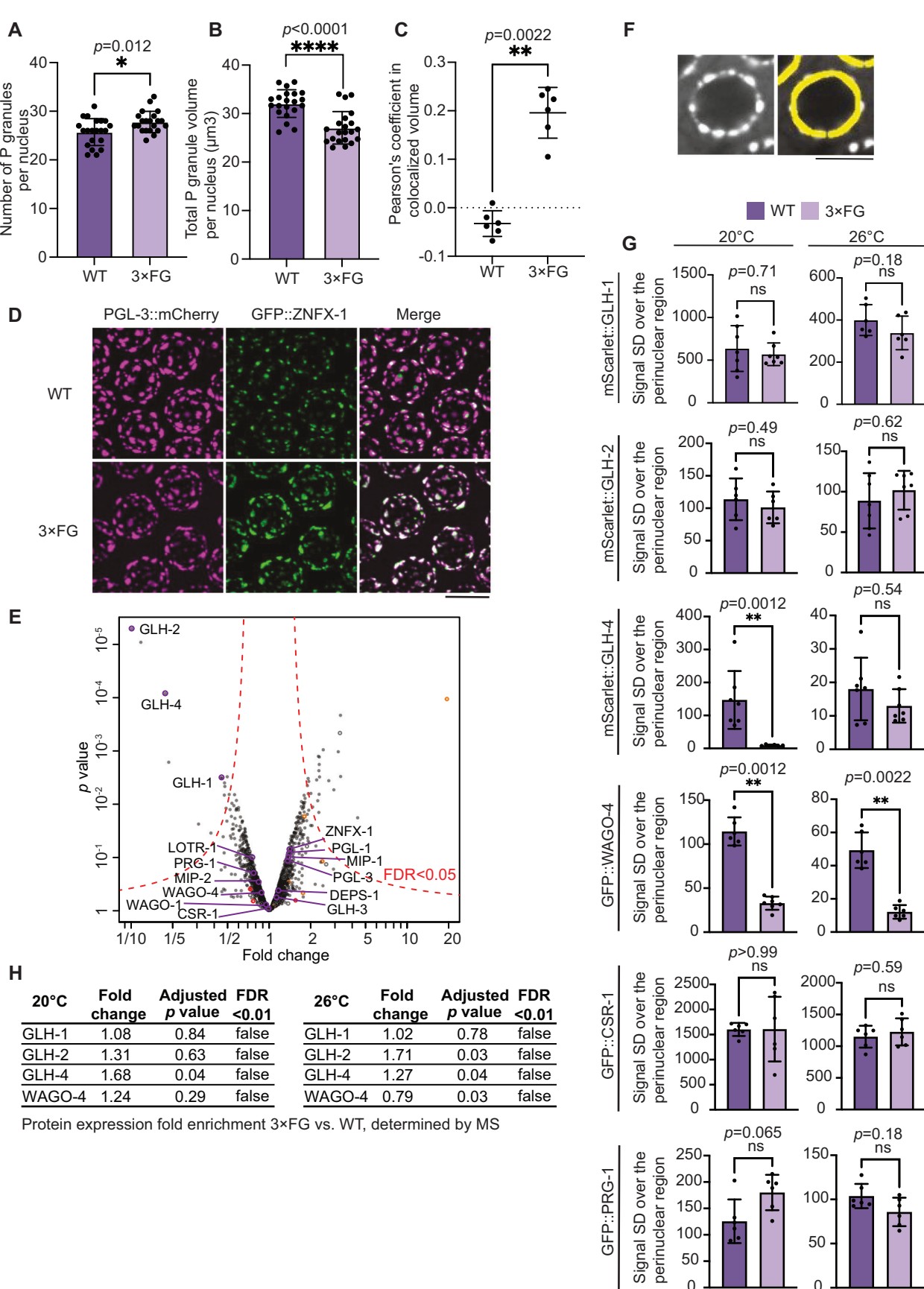

**Figure EV2.** **Impact of GLH-FG mutations on germ granule localization and composition.**

(A) Graph showing the average number of P granules per nucleus in WT and 3×FG mutant worms. Data were obtained from 3D representations of pachytene nuclei in live worms expressing PGL-3::mCherry at 20 °C. Each dot represents the number of P granules in a single nucleus ($n = 21$). Bars indicate mean values, and error bars represent SD. For statistical analysis, two-tailed Mann–Whitney test was performed, $P$ value indicated in the graph. (B) Graph showing the total P granule volume per nucleus ($\mu m^3$) in WT and 3×FG mutant worms. Data were calculated from 3D representations of pachytene nuclei in live worms expressing PGL-3::mCherry at 20 °C. Each dot represents the total P granule volume of a single nucleus ($n = 21$). Bars indicate mean values, and error bars represent SD. For statistical analysis, two-tailed Mann–Whitney test was performed, $P$ value indicated in the graph. (C) Graph showing Pearson's coefficient for colocalized volume between PGL-3::mCherry and GFP::ZNFX-1 in WT and 3×FG mutant worms. Each dot represents the coefficient for a single worm, based on approximately 30 pachytene nuclei per image ($n = 6$). Horizontal lines represent mean values, and error bars indicate SD. For statistical analysis, two-tailed Mann–Whitney test was performed, $P$ value indicated in the graph. (D) Fluorescence micrographs showing maximum intensity projections of pachytene nuclei in live worms co-expressing PGL-3::mCherry and GFP::ZNFX-1 in WT (upper) and 3×FG mutant (lower) backgrounds, grown at 20 °C. Scale bar: 5 μm. (E) Volcano plot showing fold enrichment of proteins after streptavidin pulldown in PGL-3::TurboID expressing worms grown at 20 °C (3×FG vs. WT), determined by mass spectrometry ($n = 3$ biological replicates). Statistical significance was calculated using limma (Ritchie et al, 2015). Identified known P granule and Z granule proteins are highlighted. (F) Representative fluorescent micrographs showing a single germline nucleus in a live worm expressing a fluorescently labeled germ granule protein (left panel) and the same nucleus with the ROI over the perinuclear region indicated as a yellow line (right panel). This is a representation of the ROIs used for calculating signal standard deviation in Fig. EV2G. Scale bar: 5 μm. (G) Graphs showing the average signal standard deviation over the perinuclear region in live WT and 3×FG mutant worms expressing a single fluorescently tagged protein (from top to bottom): mScarlet::GLH-1, mScarlet::GLH-2, mScarlet::GLH-4, GFP::WAGO-4, GFP::CSR-1, and GFP::PRG-1. Quantification was performed on a single confocal slice representing the middle plane of the analyzed nucleus, as shown in Fig. EV2F (each image contained multiple pachytene nuclei). Worms were grown at 20 °C (left) or 26 °C (right). Each dot represents the average signal standard deviation for a single worm ($n = 6$), bars indicate mean values, and error bars represent SD. For statistical analysis, two-tailed Mann–Whitney test was performed, $P$ values indicated in the graph. (H) Protein levels analysis by mass spectrometry of GLH proteins and WAGO-4. Mass spectrometry was performed on worm lysates from animals grown at either 20 °C or 26 °C ($n = 3$ biological replicates). Fold changes (3×FG mutant relative to WT), corresponding adjusted $P$ values and a true/false indicator for whether the false discovery rate (FDR) is below 0.01 (as determined by limma) are shown for each protein.

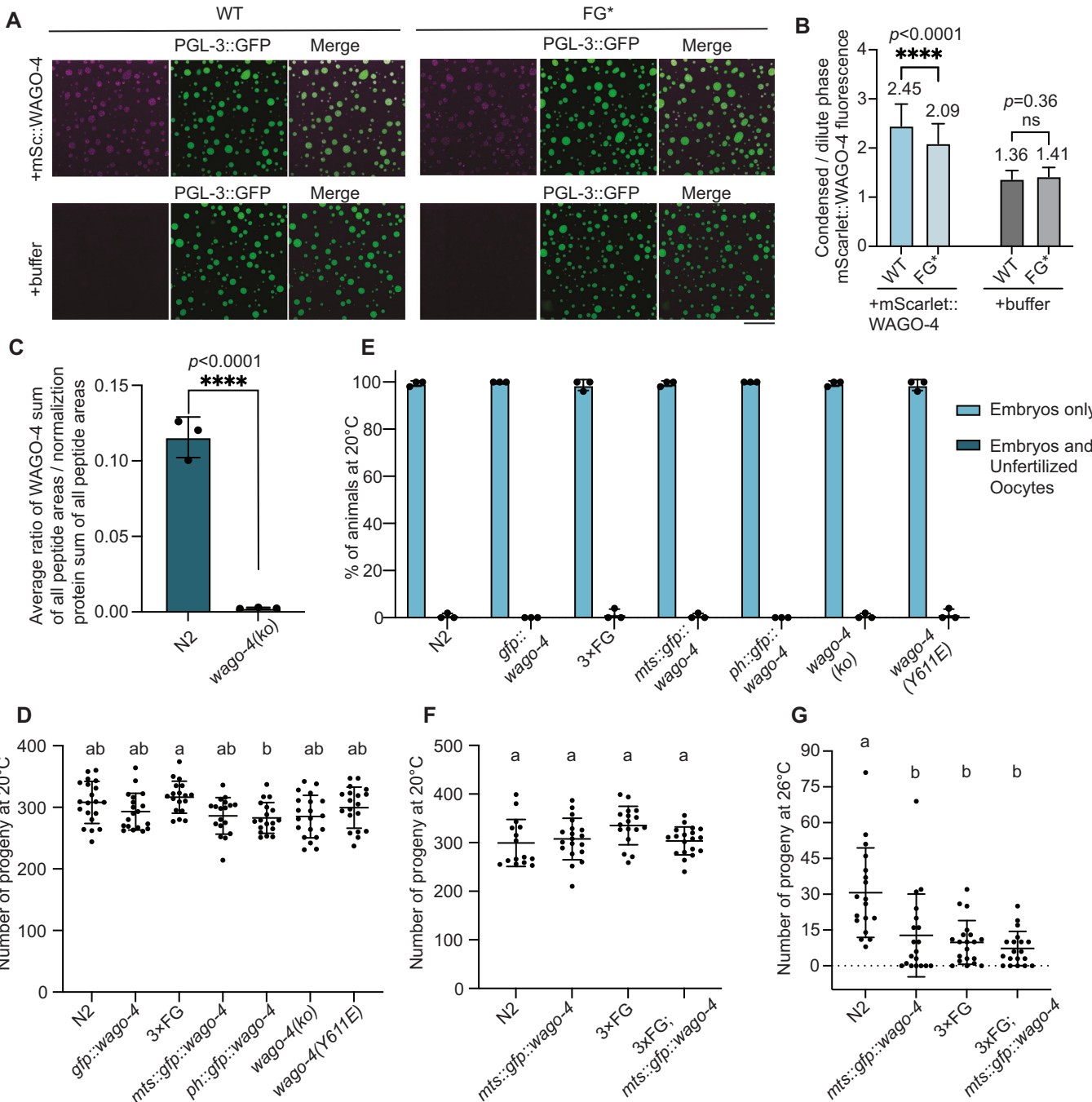

◀ **Figure EV3. WAGO-4 mislocalization from germ granules and its contribution to the 3×FG phenotype.**

(A) Fluorescent micrographs of condensates containing PGL-1, PGL-3 (5% PGL-3::mEGFP), GLH-1, and GLH-4 with the addition of mScarlet::WAGO-4 (upper row) or protein storage buffer (lower row). Condensates contained WT GLH-1 and GLH-4 (left) or F → A mutant GLH-1 and GLH-4 (right, FG*). Each image shows a single confocal plane. Scale bar: 40 µm. (B) Second replicate of the experiment shown in Fig. 3D. Graph showing the ratio of mScarlet::WAGO-4 signal intensity in the condensed phase vs. the dilute phase for WT and FG mutant condensates. Six ROIs were defined per image, and six images were analyzed for each condition ($n = 36$). Bars represent average values with SD. Numbers above bars indicate average values. Statistical analysis was performed using a two-tailed Mann–Whitney test, $P$ values indicated in the graph. mScarlet::WAGO-4 storage buffer was used as a negative control, and ratios were calculated in the same way. (C) Graph showing results from targeted PRM-MS analysis of WAGO-4 protein levels in *wago-4(ko)* strain where a stop codon was introduced into the second exon of *wago-4*. The average ratio of the sum of all WAGO-4 peptide areas vs. the sum of all peptide areas for five normalization proteins (AHCY-1, RPL-36, RPL-22, EFT-3, and RPL-27) is shown. N2 was used as a WT control. Statistical analysis was performed using a one-tailed Welch's $t$ test on $\log_2$-transformed intensity ratios, $P$ value indicated in the graph ($n = 3$ biological replicates). (D) Graph representing the number of progeny for worms of the indicated genotypes grown at 20 °C in a single experiment. Each dot represents the progeny count of a single worm ($n = 18$–20). Horizontal lines represent mean values, and error bars indicate SD. Statistical analysis: the Kruskal–Wallis test; the corresponding $P$ values are reported in Dataset EV2. Groups sharing at least one letter are statistically indistinguishable ($P < 0.05$). (E) Quantification of unfertilized oocytes in worms grown at 20 °C. Each worm was categorized into one of three groups: (1) containing exclusively embryos, (2) containing at least one embryo and at least one unfertilized oocyte, or (3) containing exclusively unfertilized oocytes (group 3 was not observed in this experiment). Data represent averages from three independent experiments ($n = 50$ worms/experiment), with SD indicated by error bars. (F) Graph representing the number of progeny for worms of the indicated genotypes grown at 20 °C. Each dot represents the progeny count of a single worm ($n = 15$–19). Horizontal lines represent mean values, and error bars indicate SD. Statistical analysis: Kruskal–Wallis test; the corresponding $P$ values are reported in Dataset EV2. Groups sharing at least one letter are statistically indistinguishable ($P < 0.05$). (G) Graph representing the number of progeny for worms of the indicated genotypes grown at 26 °C. Each dot represents the progeny count of a single worm ($n = 18$–20). Horizontal lines represent mean values, and error bars indicate SD. Statistical analysis: Kruskal–Wallis test; the corresponding $P$ values are reported in Dataset EV2. Groups sharing at least one letter are statistically indistinguishable ($P < 0.05$).

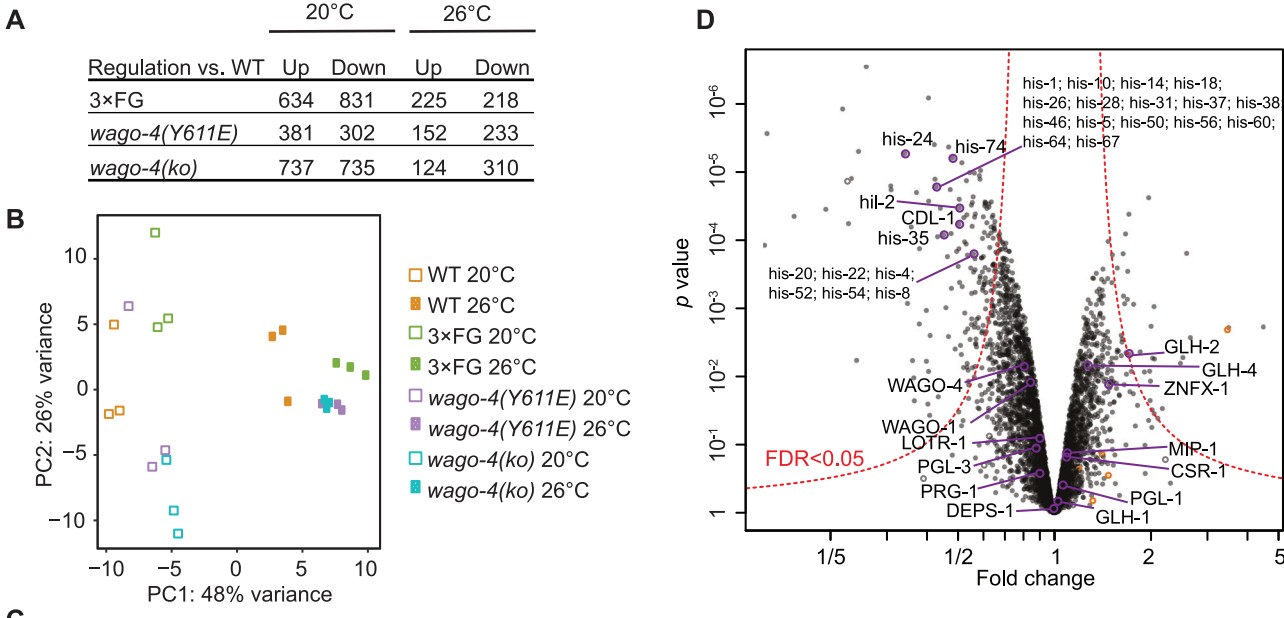

**A**

| Regulation vs. WT | 20°C | | 26°C | |
|---|---|---|---|---|
| | Up | Down | Up | Down |
| 3×FG | 634 | 831 | 225 | 218 |
| *wago-4(Y611E)* | 381 | 302 | 152 | 233 |
| *wago-4(ko)* | 737 | 735 | 124 | 310 |

**B** — PCA plot

Legend:
- WT 20°C
- WT 26°C
- 3×FG 20°C
- 3×FG 26°C
- *wago-4(Y611E)* 20°C
- *wago-4(Y611E)* 26°C
- *wago-4(ko)* 20°C
- *wago-4(ko)* 26°C

**D** — Volcano plot

**C**

| Histone/gene | hil-5 | his-1 | his-16 | his-17 | his-18 | his-2 | his-3 | his-4 | his-40 | **his-48** | his-5 | **his-59** | his-60 | **his-61** | **his-62** | **his-63** | his-64 | his-67 | his-68 | his-7 | **his-72** | his-74 |
|---|---|---|---|---|---|---|---|---|---|---|---|---|---|---|---|---|---|---|---|---|---|---|
| H1 | • | | | | | | | | | | | | | | | | | | | • | | |
| H2A | | | | • | | | • | | | | | | | | | | • | | | | • | • |
| H2B | | | | | | | | • | | | | | | | | | | | | | | |
| H3 | | | | | • | | • | | • | | | | | | | | | • | | | | |
| H4 | | • | | • | | | | | | | • | | • | | | | | • | • | | | |
| putative H2B | | | | | | | | | | • | | | | • | | | | | | | | |
| H3.3 | | | | | | | | | | | | | | | | | | | | | • | • |

**Figure EV4. Temperature-dependent transcriptomic changes in GLH-FG and WAGO-4 mutants and histone gene downregulation in 3×FG mutants.**

(A) Table showing the number of up- and downregulated genes in 3×FG, *wago-4(Y611E)*, and *wago-4(ko)* worms grown at 20 °C or 26 °C ($p < 0.05$). (B) PCA plot based on the top 500 most variable genes. Three biological replicates were analyzed for each genotype and temperature condition, with each replicate individually plotted. The legend on the right indicates the symbol for each condition. (C) Table listing all histone genes downregulated in 3×FG vs. WT based on mRNA sequencing at 20 °C. Each column represents a histone gene, and rows indicate the histone type it encodes. Genes highlighted in bold also showed increase in WAGO-4-associated 22G-RNAs in 3×FG compared to WT (Fig. EV5C). (D) Volcano plot showing fold enrichment of proteins from total worm lysates of worms grown at 26 °C (3×FG vs. WT), determined by mass spectrometry ($n = 3$ biological replicates). Statistical significance was calculated using limma (Ritchie et al, 2015). Detected histone proteins or clusters and known germ granule proteins are highlighted.

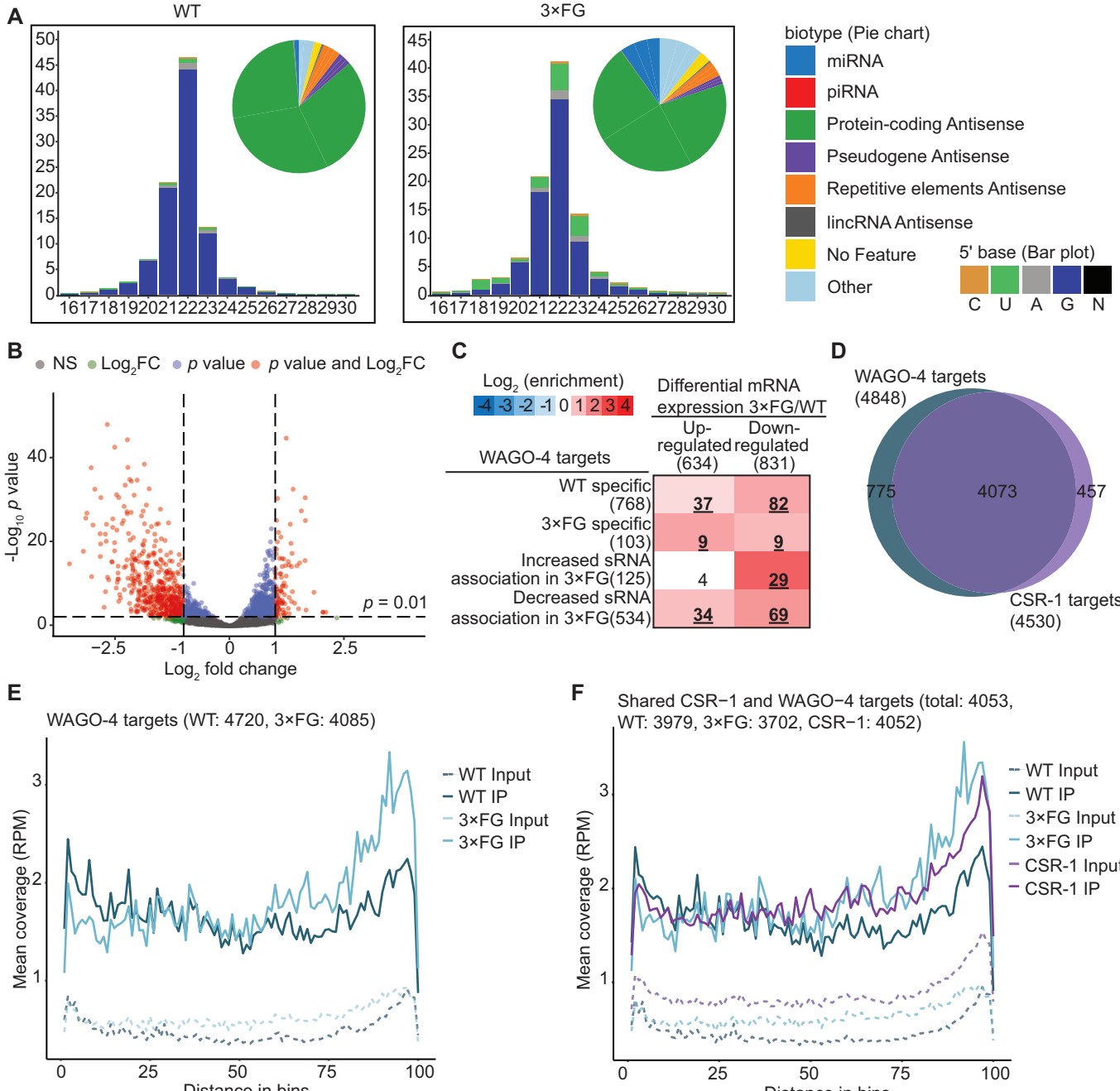

**Figure EV5.  Analysis of WAGO-4–associated 22 G RNAs and target specificity in 3×FG mutants.**

(A) Bar plots showing the 5′ nucleotide composition and length distribution of sRNAs in each Argonaute IP. Pie charts depict the proportion of sRNAs corresponding to each genetic element (biotype). Data represent the average of three biological replicates. IPs were performed using 3xFLAG::GFP::WAGO-4 in young adult worms. (B) Volcano plot showing $Log_2$ fold enrichment of 22G-RNAs targeting WAGO-4 targets in 3×FG vs. WT worms grown at 20 °C. Each dot represents the average $Log_2$ fold change ($n = 3$ biological replicates). Differential expression and statistical analysis were performed using DESeq2 (Love et al, 2014). Targets with an average $Log_2$ fold change of $<-1$ or $>1$ and $P < 0.01$ were considered to have a significant decrease or increase in WAGO-4-associated 22G-RNAs, respectively (indicated by red dots). (C) Enrichment of WAGO-4 sRNA targets in four groups: WT-specific targets, 3×FG-specific targets, targets with a decrease in WAGO-4-associated sRNAs in 3×FG, and targets with an increase in WAGO-4-associated sRNAs in 3×FG. These groups were compared to genes up- or downregulated in 3×FG/WT based on mRNA sequencing. All data represent worms grown at 20 °C. $Log_2$ enrichment is indicated by color according to the scale shown, and significant enrichment or depletion ($P < 0.05$, Fisher's exact test) is highlighted in bold and underlined. (D) Venn diagram showing the overlap of WAGO-4 targets in WT worms (this study) with CSR-1 targets (Seroussi et al, 2023). Both datasets were derived from sRNA sequencing after Ago IP from adult worms grown at 20 °C. Numbers in brackets indicate the size of each group. (E) Metagene profiles for sRNAs complementary to protein-coding gene WAGO-4 targets (all identified targets in WT and 3×FG worms grown at 20 °C). Size-normalized targets were partitioned into 100 bins, and the mean coverage (RPM) for each bin was plotted. Solid lines represent IP values, and dashed lines represent input values. The number of identified targets for each genotype is indicated in brackets. (F) Metagene profiles for sRNAs complementary to protein-coding gene WAGO-4 and CSR-1 targets. CSR-1 targets were identified from CSR-1 IP in WT worms (Seroussi et al, 2023). Size-normalized targets were partitioned into 100 bins, and mean coverage (RPM) for each bin was plotted. Solid lines represent IP values, and dashed lines represent input values. The total number of genes analyzed was determined as WAGO-4 and CSR-1 shared targets based on published data (Seroussi et al, 2023), and the number of identified targets for each genotype is indicated in brackets.

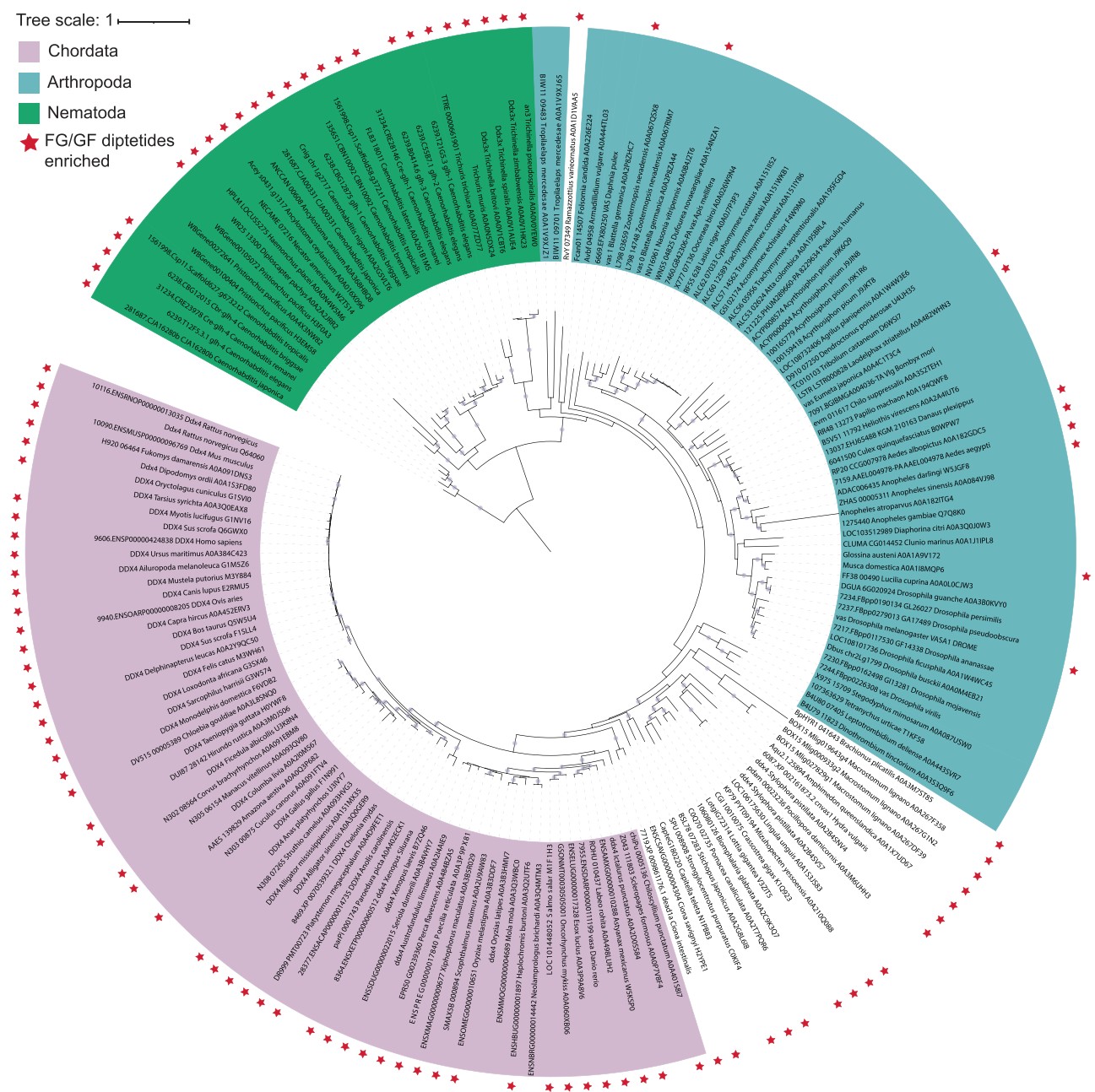

**Figure EV6. Phylogenetic tree of Vasa orthologs.**

Branches that are supported by an ultrafast bootstrap (UFBoot) value >=95% are indicated by a gray dot. Branch lengths represent the inferred number of amino acid substitutions per site, and branch labels are composed of gene name (if available), genus, species, and accession number. Those proteins which have significant enrichment of FG or GF dipetides enriched in their sequence (excluding the helicase domains) are highlighted with red stars ($P < 0.005$).

