## [Peer Review File · The EMBO Journal]

Germ Granule Localization of nematode Argonaute WAGO-4 Ensures Fidelity in Small RNA Loading

Stela Jelenic, Mathias Renaud, Samantha Del Borrello, Joseph Gokcezade, Janos Bindics, Lisa Frasz, Philipp Czermak, Peter Duchek, and Julie Claycomb

Corresponding author: Stela Jelenic (stela.jelenic@imba.oeaw.ac.at)

Review Timeline:

Submission Date:	15th Apr 25
Editorial Decision:	9th May 25
Revision Received:	6th Aug 25
Editorial Decision:	22nd Aug 25
Revision Received:	5th Sep 25
Accepted:	14th Sep 25

Editor: Yehu Moran

Transaction Report:

Dear Dr. Jelenic,

Thank you for submitting your manuscript for consideration by the EMBO Journal. It has now been seen by three referees whose comments are shown below.

Given the referees' positive recommendations, I would like to invite you to submit a revised version of the manuscript, addressing the comments of all three reviewers. I should add that it is EMBO Journal policy to allow only a single round of revision, and acceptance of your manuscript will therefore depend on the completeness of your responses in this revised version.

I suggest that in order to increase the chances of acceptance for your manuscript and for making the next steps of the review smoother, you will prepare a revision plan together with your co-authors and send it to me within a few weeks via email. From our past experience this can make a difference and save you and your co-authors significant time and effort as this will help in coordinating expectations and potentially allow us to provide you with feedback regarding your revision plan.

Thank you for the opportunity to consider your work for publication. I look forward to your revision.

Sincerely,

Yehu Moran
Academic Editor
The EMBO Journal

We realize that it is difficult to revise to a specific deadline. In the interest of protecting the conceptual advance provided by the work, we recommend a revision within 3 months (7th Aug 2025). Please discuss the revision progress ahead of this time with the editor if you require more time to complete the revisions.

Referee #1:

In this manuscript, the authors investigate how disrupting the localization of the Argonaute WAGO-4 to germ granules affects fertility and gene regulation. They show that mutations in the FG repeats of several Vasa-like GLH proteins cause WAGO-4 to mislocalize to the cytoplasm instead of concentrating in germ granules, as seen in wild-type animals. This cytoplasmic localization of WAGO-4 leads to reduced fertility at (highly) elevated temperatures. Additionally, transcriptome and WAGO-4-bound small RNA analyses in GLH mutant worms reveal that WAGO-4 mislocalization alters its regulatory activity. In particular, WAGO-4's ability to target genes not co-regulated with CSR-1 is impaired. These findings underscore the critical role of germ granules in controlling Argonaute function and maintaining proper gene regulation.

Major comments:

1. Testing fertility at 26{degree sign}C is somewhat problematic, as this temperature is near the threshold for complete sterility in *C. elegans*, and the presented results show considerable variability. I recommend to also assess the fertility of FG mutants, as well as the *wago-4* alleles (MTS, PH, Y611E, and the KO) at 25{degree sign}C. While the sequencing data at 20{degree sign}C are compelling, it could be beneficial - though optional - to also perform mRNA and sRNA sequencing at 25{degree sign}C for better alignment with the phenotype.
2. Based on the data, it seems plausible that the GLH-4(F→A) mutation alone may be sufficient to cause both the fertility defects and the mislocalization of WAGO-4. Testing the GLH-4 single mutant could help narrow down the specific contribution of each GLH protein to the observed phenotypes.
3. The authors propose that GLH FG mutations cause fertility defects partially by inducing WAGO-4 mislocalization (lines 249-251), and support this by showing that direct mislocalization of WAGO-4 via specific *wago-4* mutants also leads to reduced fertility. To further support this model, I recommend testing worms with combined GLH FG mutations with a *wago-4* mislocalization mutation. If no additive effect is observed, it would strengthen the argument that both act through the same pathway, consistent with the proposed mechanism.
4. It would be helpful to examine the expression levels of *wago-4* in the mRNA-seq data. If *wago-4* expression is unchanged in the FG mutants, this should be noted, as it may suggest that the cytoplasmic function of WAGO-4 is over expressed in these mutants.
5. The authors suggest that fewer gene expression changes are observed at 26{degree sign}C due to stress responses masking more subtle regulatory effects. To support this, I would like to see a comparison of stress-related gene expressions between the different temperatures (for both genotypes). Additionally, an alternative explanation could be germline degeneration at 26{degree sign}C - reduced germline content may limit the ability to detect WAGO-4-related changes, leading to an apparent reduction in differential expression between genotypes (sequencing from 25{degree sign}C may solve this problem).

Minor comments:

1. Line 38 - add the word epigenetic after the word transgenerational.
2. Line 50 - move the acronym (sRNA) to line 40 after the first use of the term small RNA.
3. line 257 should be (Fig. 3H) and not (Fig. 4H).
4. line 355 says 426 of 768 but in the figure, it is 392 and not 426.
5. line 361 says 117/125 but the figure says 113.
6. It is not clear whether the results shown from the sRNA sequencing are from worms grown at 20{degree sign}C or 26{degree sign}C.

Referee #2:

WAGO-4 germ granule localization ensures fidelity in small RNA loading

In this study, the authors assigned a functional role to germ granule localization of WAGO-4. The germ granule proteins GLH-1, -2 and -4 contain FG repeats that may form a dense meshwork and a diffusion barrier, which is well known from the NPC. The authors reasoned that these repeats are responsible for germ granule formation and deleted them from the three GLH proteins. A phenotype on oocyte maturation is overserved at elevated temperatures. GLH proteins are found in P granules, and such granules can still form when the FG repeats are deleted. However, the composition is altered. Among other proteins, WAGO-4 is reduced in such granules. Artificial mis-localization of WAGO-4 resulted in similar phenotypes as observed for the FG deletion mutants. The FG repeat mutants show also an altered mRNA expression program, which correlates with WAGO-4 targets suggesting a role for the FG repeats in WAGO-4 function. Finally, the authors investigated small RNAs associated with WAGO-4 in wt and FG mutants. When the FG repeats are deleted, WAGO-4 carries more small RNAs targeting RNAs that are shared with Csr1 suggesting less granule association and more loading of soluble cytoplasmic small RNAs into WAGO-4.

This is a clear and competently performed study that addresses an unsolved and interesting topic. It is unclear how much granule localization contributes to the function of Argonaute proteins in general and the study contributes to this aspect of gene silencing. There are, however, a number of weaker and unclear aspects that are listed below.

1. The authors hypothesized that FG repeats may form interaction networks potentially resulting in granule formation. Similar effects are observed in nuclear pores, for example. However, granule formation seems to be independent of the FG repeats of GLH1, 2 and 4 since they still form in the mutants. A more direct role of FG repeats in interacting with proteins such as WAGO-4 is therefore more likely (as the authors mention). Do these repeats interact with WAGO-4? This could be directly tested.
2. Are the GLH levels in wt and mutant worms similar? This could be tested by western blotting and quantification.
3. Similarly, are WAGO-4 levels changed in FG repeat mutant worms?
4. Mis-localizing WAGO-4 to exclude it from P granules is certainly a straight-forward strategy. A problematic aspect could be that unpredictable effects could contribute to the phenotype (gain-of-function). WAGO-4 is not only absent from P granules but also present at places where it is normally not found. These results should be treated with caution and not be over interpreted.
5. An example image of WAGO-4-associated in vitro produced condensates could be included.
6. Since there are a number of proteins mis-localized upon FG repeat deletion, the observed phenotype might be more complex and cannot simply be associated with WAGO-4 absence from P granules. Maybe this could be made clearer throughout the text.

Referee #3:

Germ granules are biomolecular condensates that play critical roles in germ cell development and fidelity through small RNA regulatory pathways. Argonautes are critical players within germ granules and these RNA regulatory pathways, but the connection between proper granule formation and Argonaute function remains an important biological question. Here, Jelenic, et al. study the requirement of intrinsically disordered regions for Argonaute WAGO-4 assembly and function in P granules. Mutation and/or deletion of the FG repeats of the GLH Vasa helicases lead to decreased fecundity at higher temperatures, despite apparent germ granule formation. PGL-3 BioID identified association with WAGO-4 protein, and note that the GLH FG mutants (3xFG) cause mislocalization of WAGO-4 from P and Z granules. This mislocalization was also modestly observed in an in vitro granule assembly assay. Modification of WAGO-4 to mislocalize it from germ granules also affected fecundity and caused a mortal germline phenotype. mRNA-seq in mutant worm strains revealed differences between the mutants and WAGO-4 knockout worms. WAGO-4 small RNA-seq showed evidence that the Argonaute loses the robust ability to bind to 22G-RNAs antisense to WAGO-4 specific, non- CSR-1 Argonaute, RNA targets. The manuscript proposes a model where mislocalization from germ granules changes the loading dynamics of WAGO-4.

This manuscript is clearly written and well cited. I appreciate that it potentially (see Major comments) identifies a mutant protein responsible for a specific phenotype, rather than a null mutation that disrupts the entire system. However, the genomic findings are not entirely definitive, potentially due to the delicate molecular balance between CSR-1 and WAGO-4 that may result in only subtle effects. The study would benefit from some insight into how WAGO-4's molecular function might be impacted by its mislocalization. Nonetheless, the work builds on previous studies and will be of interest to researchers focused on granule structure and function, Argonaute proteins, and RNA and developmental biology more generally.

MAJOR

1. Figure 1B; 2F; S1C,E,J: Recommend reporting complete statistics between a and b groups.
2. Figure 1D; 2G; S1H,I: Recommend reporting statistics between strains and categories.

3. Figure 2H: Please report statistics between N2 and mutant worms.
4. Total GLH mutant proteins levels appear different than wild type protein in Fig 2. Were total GLH protein levels assessed at different temperatures? GLH protein levels may be a cause of fecundity differences between strains.
5. The mRNA- and WAGO-4 sRNA-seq assays greatly complement the mutant study, but the results are somewhat convoluted and descriptive. The manuscript will be greatly strengthened with a molecular readout assay for WAGO-4 function. Were RNAi inheritance assays attempted with the WAGO FG and mislocalization mutants (e.g. Xu et al. 2018 (DOI: 10.1016/j.celrep.2018.04.072))? The results may help further support granule localization as a site required for RNAi exogenous inheritance; if FG/localization mutant results mimics wago-4 null worms, this will support the granule as the subcellular location for WAGO-4 exogenous RNAi. I realize this moves away from the manuscript's intent on investigating WAGO-4's function in regulating endogenous genes, but it does support the model of WAGO-4 using two distinct sRNA loading pathways for exogenous and endogenous gene targets, as described in Figure 6.
6. Figure 6: "Our findings support a model (Fig. 6) in which WAGO-4 utilizes at least two distinct small RNA loading pathways: a cytoplasmic pathway shared with CSR-1 and a germ granule-dependent pathway required for targeting unique transcripts." This WAGO-4 sRNA loading model is not clearly depicted in the Figure.
7. Recommend adding a statistics sentence/paragraph to the methods or statistics description in the appropriate sections to mention the software used for calculations.
8. A brief description of the CRISPR-Cas strategy used should be described in the methods. Were the fluorescent and bioID genes amplified from a plasmid? It is also not clear to me whether the GLH F to A mutations were each made individually or gene fragment blocks inserted.

MINOR

1. Figure 5B: The bold numbers are difficult to distinguish from the non-bold numbers. Recommend revision for clarity.
2. In an ideal world, the representative images from Fig S3A,C would be presented in the main Figure prior to the quantitation. Consider moving (a) the images to the main figure +/- splitting up the Figure to accommodate space, or (b) moving the in vitro data to the supplement and the germline images to the main figure as this is the more striking result.

We would like to express our thanks for the thoughtful and constructive feedback provided by the editor and reviewers. We believe that addressing the insights shared by reviewers has improved the clarity, quality, and overall impact of our study. Below, we provide point-by-point responses to the referees' comments (in blue).

Referee #1:

In this manuscript, the authors investigate how disrupting the localization of the Argonaute WAGO-4 to germ granules affects fertility and gene regulation. They show that mutations in the FG repeats of several Vasa-like GLH proteins cause WAGO-4 to mislocalize to the cytoplasm instead of concentrating in germ granules, as seen in wild-type animals. This cytoplasmic localization of WAGO-4 leads to reduced fertility at (highly) elevated temperatures. Additionally, transcriptome and WAGO-4-bound small RNA analyses in GLH mutant worms reveal that WAGO-4 mislocalization alters its regulatory activity. In particular, WAGO-4's ability to target genes not co-regulated with CSR-1 is impaired. These findings underscore the critical role of germ granules in controlling Argonaute function and maintaining proper gene regulation.

We thank Referee #1 for their positive comments about the study and the detailed feedback, which has been very helpful in strengthening our manuscript.

Major comments:

1. Testing fertility at 26°C is somewhat problematic, as this temperature is near the threshold for complete sterility in *C. elegans*, and the presented results show considerable variability. I recommend to also assess the fertility of FG mutants, as well as the *wago-4* alleles (MTS, PH, Y611E, and the KO) at 25°C. While the sequencing data at 20°C are compelling, it could be beneficial - though optional - to also perform mRNA and sRNA sequencing at 25°C for better alignment with the phenotype.

We appreciate this point and agree that 26°C is a high temperature for *C. elegans*. We previously tested fecundity of *glh* FG mutants at 25°C and found that both 2xFG and 3xFG mutants produced significantly fewer progeny than wild-type animals. Although the phenotype was apparent at 25°C, we chose to conduct subsequent experiments at 26°C because the phenotype was more pronounced at this temperature, and this temperature has commonly been used in the literature for assessing *glh* mutants (Marnik et al., 2019; Spike et al., 2008). To address the referee's concern, we included the 25°C fecundity data in the revised manuscript, demonstrating that the phenotype is detectable under milder stress conditions (Line 131):

“As 26°C is a high temperature for C. elegans, we also tested the fecundity of FG mutants at 25°C and observed that both 2×FG and 3×FG mutants produced significantly fewer progeny compared to wild-type animals (Fig. EV1D). Although the phenotype was already apparent at 25°C, we conducted subsequent experiments at 26°C because (a) the phenotype was more pronounced at this temperature, and (b) 26°C has been commonly used in the literature to assess fertility in glh mutants (e.g., Marnik et al, 2019; Spike et al, 2008).”

Regarding the suggestion to perform mRNA and sRNA sequencing at 25°C, we appreciate the referee’s perspective. Given the time, cost, and considering that our sequencing analysis was conducted at 20°C (avoiding the potentially problematic 26°C temperature, yet still revealing molecular phenotypes), we decided not to pursue this analysis at 25°C.

2. Based on the data, it seems plausible that the GLH-4(F→A) mutation alone may be sufficient to cause both the fertility defects and the mislocalization of WAGO-4. Testing the GLH-4 single mutant could help narrow down the specific contribution of each GLH protein to the observed phenotypes.

To address this possibility, we conducted fecundity assays for each single FG mutant (*glh-1*, *glh-2*, and *glh-4*) at 26°C, using the N2 strain as a negative control and the 2xFG (*glh-1 glh-4* double mutant) as a positive control. Our results showed that at 26°C, all the single FG mutants had similar progeny numbers compared to the N2 control, while the double *glh-1 glh-4* mutant produced significantly fewer progeny. We have included these results in the manuscript (line 137):

*“We tested all the single FG mutants (*glh-1*(F→A), *glh-2*(ΔFG) and *glh-4*(F→A)) to identify the contribution of each GLH protein FG domain to the observed fecundity phenotype at 26°C. Our results showed that all single FG mutants exhibited similar progeny numbers to the N2 control (Fig. EV1E), indicating that mutations in multiple FG domains are required to observe the phenotype.”*

We also assessed WAGO-4 localization in the *glh-4*(F→A) single mutant and found no significant difference compared to the wild-type. This result is consistent with the lack of a fecundity phenotype in the *glh-4*(F→A) single mutant. While we found this suggestion valuable and important to address, we encountered some difficulty integrating this result into the current manuscript, as WAGO-4 localization is introduced later in the text. Nonetheless, we appreciate the referee’s suggestion and provided the results here:

Left: Representative fluorescence micrographs of gonads (pachytene region) from live worms expressing GFP::WAGO-4. A single confocal slice is shown for worms with WT and *glh-4(F→A)* mutant backgrounds grown at 20°C and 26°C. Scale bar: 5 μm.

Right: Graphs showing the average signal standard deviation over the perinuclear region in live WT and *glh-4(F→A)* mutant worms expressing GFP::WAGO-4. Quantification was performed on a single confocal slice representing the middle plane of the analyzed nucleus, as shown in Fig. EV2F (each image contained multiple pachytene nuclei; n=6 worms). Worms were grown at 20°C (left) or 26°C (right). Each dot represents the average signal standard deviation for a single worm (n=6), bars indicate mean values, and error bars represent SD. For statistical analysis, Mann-Whitney test was performed ($p=0.065$ for 20°C and $p=0.59$ for 26°C).

3. The authors propose that GLH FG mutations cause fertility defects partially by inducing WAGO-4 mislocalization (lines 249-251), and support this by showing that direct mislocalization of WAGO-4 via specific *wago-4* mutants also leads to reduced fertility. To further support this model, I recommend testing worms with combined GLH FG mutations with a WAGO-4 mislocalization mutation. If no additive effect is observed, it would strengthen the argument that both act through the same pathway, consistent with the proposed mechanism.

We agree that testing the genetic interaction would provide additional insights. We have generated a double mutant combining the 3xFG background with the *mts::wago-4* mislocalization mutant. Our fecundity tests did not show an additive effect at either 20°C or 26°C, suggesting that the GLH FG mutations and WAGO-4 mislocalization likely act through the same pathway. We have included these results in the manuscript (line 288):

“Additionally, we have investigated the genetic interaction between FG mutations in GLH proteins and WAGO-4 mislocalization. We generated a double mutant combining the 3xFG background with the mts::gfp::wago-4 mislocalization mutant. Our fecundity tests revealed no additive effect at either 20°C or 26°C (Fig. EV3F, G). This suggests that FG mutations in GLH proteins and WAGO-4 mislocalization likely act through the same pathway.”

4. It would be helpful to examine the expression levels of *wago-4* in the mRNA-seq data. If *wago-4* expression is unchanged in the FG mutants, this should be noted, as it may suggest that the cytoplasmic function of WAGO-4 is over expressed in these mutants.

This is an important point made by all 3 referees (R2 comments 2 and 3, R3 comment 4).

We agree that knowing whether the expression level of WAGO-4 changes in the mutants is important for interpreting our results. We performed mass spectrometry to assess protein levels in wild type vs. 3xFG mutants. These data show that the levels of WAGO-4, as well as the GLHs (related to R2's and R3's queries) in the 3xFG mutants are comparable to those in wild-type animals. These data have been added to the revised manuscript (line 227):

“To evaluate whether differences in the protein levels of WAGO-4 (or GLH proteins) might contribute to the phenotype, we measured their abundance in the 3xFG mutants using mass spectrometry. Our analysis revealed that the levels of WAGO-4, as well as

GLH-1, GLH-2 and GLH-4, in the 3×FG mutants are comparable to those in wild-type animals (Fig. EV2H)."

We also appreciate the referee's point regarding the possibility of increased cytoplasmic concentration of WAGO-4, due to altered localization in the 3×FG mutants (even without unaltered expression levels). We also recognize that the forced localization to ectopic sites could lead to gain-of-function effects influencing the phenotype. We have included this information in the revised manuscript (Line 294):

"Together, these findings demonstrate that wago-4 mutants and 3×FG mutants exhibit similar germline phenotypes and that the FG mutations in GLH proteins and WAGO-4 mislocalization likely impair fertility through the same pathway. We cannot entirely exclude unintended gain-of-function effects resulting from WAGO-4 mislocalization to ectopic sites (cytoplasm, mitochondria, plasma membrane) and we acknowledge that GLH-4 and potentially other granule proteins are also mislocalized in the 3×FG mutant, which could lead to a complex phenotype. However, the consistency of phenotypes across the diverse WAGO-4 mislocalization contexts and the loss-of-function mutant, and similarity with 3×FG mutant phenotypes suggest that impaired WAGO-4 granule localization is likely the primary contributor to the observed fertility defects. This indicates that WAGO-4 function is compromised when its normal enrichment in germ granules is disrupted."

5. The authors suggest that fewer gene expression changes are observed at 26°C due to stress responses masking more subtle regulatory effects. To support this, I would like to see a comparison of stress-related gene expressions between the different temperatures (for both genotypes). Additionally, an alternative explanation could be germline degeneration at 26°C - reduced germline content may limit the ability to detect WAGO-4-related changes, leading to an apparent reduction in differential expression between genotypes (sequencing from 25°C may solve this problem).

We appreciate the suggestion to examine stress-related gene expression as a potential factor influencing the reduced number of identified differentially expressed genes at 26 °C. As we speculated in the manuscript, stress responses might mask more subtle regulatory effects. To explore this possibility further, we performed an analysis of stress-related gene expression using our existing mRNA sequencing data to compare the same genotypes at 20°C and 26°C. We have included the results in the revised manuscript (Line 326):

"We speculate that worms experience substantial stress at 26°C, triggering a generalized stress response in both WT and mutant animals, potentially masking more subtle differences in expression between genotypes. Supporting this hypothesis, we

observed greater gene expression variation between 26°C and 20°C within the same strain (WT: 2,309 differentially regulated genes, 3×FG: 1,431, wago-4(Y611E): 2,281, wago-4(ko): 2,614; data provided in Table EV2) than between WT and mutants at 26°C (3×FG: 443 differentially regulated genes, wago-4(Y611E):385, wago-4(ko): 434; Fig. EV4A). Additionally, analysis of the genes commonly regulated at 26°C across all strains revealed significant enrichment for stress response genes among the up-regulated genes ($p=1.39e-17$). For the down-regulated genes, ribosome biogenesis and translation initiation factors were most enriched ($p=1.39e-51$) (data provided in Table EV2). This pattern aligns with the reported global decrease in translation under stress to conserve energy for essential processes and stress adaptation (Holcik and Sonenberg, 2005).”

Minor comments:

1. Line 38 - add the word epigenetic after the word transgenerational.
2. Line 50 - move the acronym (sRNA) to line 40 after the first use of the term small RNA.
3. line 257 should be (Fig. 3H) and not (Fig. 4H).
4. line 355 says 426 of 768 but in the figure, it is 392 and not 426.
5. line 361 says 117/125 but the figure says 113.
6. It is not clear whether the results shown from the sRNA sequencing are from worms grown at 20{degree sign}C or 26{degree sign}C.

We have corrected these typos and clarified that the small RNA sequencing analysis was performed on worms grown at 20°C. This is now explicitly stated in the revised manuscript (Line 376):

“To investigate how WAGO-4 function is affected in the 3×FG mutant, we assessed the repertoire of sRNAs associated with WAGO-4 and compared it to WT. We performed sRNA sequencing coupled with immunoprecipitation (IP) of WAGO-4 (Fig. EV5A) from samples of worms grown at 20°C.”

Referee #2:

Jelenic et al.

WAGO-4 germ granule localization ensures fidelity in small RNA loading

In this study, the authors assigned a functional role to germ granule localization of WAGO-4. The germ granule proteins GLH-1, -2 and -4 contain FG repeats that may

form a dense meshwork and a diffusion barrier, which is well known from the NPC. The authors reasoned that these repeats are responsible for germ granule formation and deleted them from the three GLH proteins. A phenotype on oocyte maturation is observed at elevated temperatures. GLH proteins are found in P granules, and such granules can still form when the FG repeats are deleted. However, the composition is altered. Among other proteins, WAGO-4 is reduced in such granules. Artificial mis-localization of WAGO-4 resulted in similar phenotypes as observed for the FG deletion mutants. The FG repeat mutants show also an altered mRNA expression program, which correlates with WAGO-4 targets suggesting a role for the FG repeats in WAGO-4 function. Finally, the authors investigated small RNAs associated with WAGO-4 in wt and FG mutants. When the FG repeats are deleted, WAGO-4 carries more small RNAs targeting RNAs that are shared with Csr1 suggesting less granule association and more loading of soluble cytoplasmic small RNAs into WAGO-4.

This is a clear and competently performed study that addresses an unsolved and interesting topic. It is unclear how much granule localization contributes to the function of Argonaute proteins in general and the study contributes to this aspect of gene silencing. There are, however, a number of weaker and unclear aspects that are listed below.

We thank Referee #2 for their thoughtful and constructive comments, and for recognizing the relevance and novelty of our study. We appreciate the opportunity to clarify several points and have revised the manuscript accordingly to address the specific issues noted below.

1. The authors hypothesized that FG repeats may form interaction networks potentially resulting in granule formation. Similar effects are observed in nuclear pores, for example. However, granule formation seems to be independent of the FG repeats of GLH1, 2 and 4 since they still form in the mutants. A more direct role of FG repeats in interacting with proteins such as WAGO-4 is therefore more likely (as the authors mention). Do these repeats interact with WAGO-4? This could be directly tested.

We appreciate the referee's suggestion to test whether the FG repeats in GLH proteins directly interact with WAGO-4. This is indeed an important and intriguing question that we have considered during the course of our study. While our data suggest that FG repeats may contribute to protein-protein interactions relevant to granule composition, we agree that granule formation itself appears to be independent of the FG domains in GLH-1, GLH-2, and GLH-4. We suspect that the FG repeats play a more subtle role, likely fine-tuning granule properties and protein composition, rather than serving as a

major driver of granule formation. We hypothesize that this fine-tuning may occur through weak, transient interactions with proteins such as WAGO-4.

We have explored several approaches to directly test the FG repeat–WAGO-4 interactions. Unfortunately, computational methods like AlphaFold are not well suited to modeling disordered regions such as FG repeats, and immunoprecipitation-based assays tend to be ineffective at detecting weak, transient interactions typical of FG domains, even in systems like nucleoporins where these motifs are more abundant. While specialized assays have been developed to probe such interactions (e.g., Patel et al., 2007), we are currently limited by the inability to purify recombinant WAGO-4, which prevents us from pursuing these experiments at this time.

At present, the most direct approach available to us is our *in vitro* reconstitution assay (Figs. 3C, D; EV3A, B), which tests whether FG repeat–dependent interactions influence WAGO-4 partitioning into granules. The results suggest that FG repeats may contribute to WAGO-4 recruitment into phase-separated granules, though we acknowledge that additional factors and active cellular processes are likely involved *in vivo*. We have revised the relevant paragraph in the Discussion section to better address the points discussed here (Line: 471)

“GLH proteins are Vasa-like DEAD-box RNA helicases essential for RNA remodeling and sRNA pathway regulation (Gruidl et al, 1996; Kuznick et al, 2000; Marnik et al, 2019; Chen et al, 2020, 2022; Dai et al, 2022). While their helicase activity is critical for sRNA pathways, FG repeats play a distinct role. FG repeats have been proposed to interact with nuclear pore complex (NPC) FG domains to anchor P granules (Updike et al, 2011), but their role in recruiting GLH proteins or other components to perinuclear granules remains unclear (Marnik et al, 2019; Chen et al, 2020). Our 3×FG mutants allowed us to dissect FG repeat function (Ribbeck and Görlich 2002; Patel et al, 2007): we found that P granules were formed and remained perinuclear in these mutants, suggesting GLH-FG repeats are not strictly required for granule formation or anchoring. Consistently, a recent study shows that efficient reduction of perinuclear P granules requires the removal of an FG-containing nucleoporin and the P granule factor MIP-1 (Thomas, Bodas, and Seydoux 2025). Our data demonstrate that FG repeats in GLH proteins are critical for proper WAGO-4 localization to germ granules, though the mechanism remains unclear. They may directly recruit WAGO-4, similar to how Drosophila GW182 recruits Ago1 to P bodies (Eulalio et al, 2009), or alternatively, fine-tune condensate composition by modulating hydrophobicity. Supporting the latter, previous work indicated that P granule integrity relies on hydrophobic interactions (Updike et al, 2011). Our in vitro reconstitution experiments suggest that FG repeats contribute to WAGO-4 recruitment into phase-separated granules, but their loss alone is not sufficient to fully abrogate WAGO-4 recruitment to granules. While further research is needed to elucidate the precise mechanism of FG repeat function in germ granules,

current data support the idea that they play a subtle role in modulating granule hydrophobicity and, consequently, protein composition.”

2. Are the GLH levels in wt and mutant worms similar? This could be tested by western blotting and quantification.

AND

3. Similarly, are WAGO-4 levels changed in FG repeat mutant worms?

Referees 1 and 3 had similar queries (R1, comment 4 and R3, comment 4), addressed above.

In short, we have examined the levels of WAGO-4 and GLH proteins in our mass spectrometry data and find that levels in the 3xFG mutants are comparable to those in wild-type animals. These data have been added to the revised manuscript (line 227):

“To evaluate whether differences in the protein levels of WAGO-4 (or GLH proteins) might contribute to the phenotype, we measured their abundance in the 3xFG mutants using mass spectrometry. Our analysis revealed that the levels of WAGO-4, as well as GLH-1, GLH-2 and GLH-4, in the 3xFG mutants are comparable to those in wild-type animals (Fig. EV2H).”

4. Mis-localizing WAGO-4 to exclude it from P granules is certainly a straight-forward strategy. A problematic aspect could be that unpredictable effects could contribute to the phenotype (gain-of-function). WAGO-4 is not only absent from P granules but also present at places where it is normally not found. These results should be treated with caution and not be over interpreted.

We appreciate the referee’s cautionary point regarding potential gain-of-function effects arising from mis-localizing WAGO-4. We fully agree with this concern. Notably, we observed the same fertility defect phenotype when WAGO-4 was mislocalized to distinct ectopic locations such as the cytoplasm, mitochondria, and plasma membrane. Given that these locations are diverse and do not share an obvious common function, the consistency of the phenotype suggests that the fertility defect results primarily from loss of proper germ granule localization, rather than a gain-of-function effect specific to a particular ectopic site. However, we agree that it is crucial to explicitly acknowledge this caveat in our manuscript to avoid over-interpretation, and we have modified the text accordingly (Line 294):

“Together, these findings demonstrate that wago-4 mutants and 3xFG mutants exhibit similar germline phenotypes and that the FG mutations in GLH proteins and WAGO-4

mislocalization likely impair fertility through the same pathway. We cannot entirely exclude unintended gain-of-function effects resulting from WAGO-4 mislocalization to ectopic sites (cytoplasm, mitochondria, plasma membrane) and we acknowledge that GLH-4 and potentially other granule proteins are also mislocalized in the 3×FG mutant, which could lead to a complex phenotype. However, the consistency of phenotypes across the diverse WAGO-4 mislocalization contexts and the loss-of-function mutant, and similarity with 3×FG mutant phenotypes suggest that impaired WAGO-4 granule localization is likely the primary contributor to the observed fertility defects. This indicates that WAGO-4 function is compromised when its normal enrichment in germ granules is disrupted.”

5. An example image of WAGO-4-associated in vitro produced condensates could be included.

An image of WAGO-4-associated condensates is shown in Figure EV3A.

6. Since there are a number of proteins mis-localized upon FG repeat deletion, the observed phenotype might be more complex and cannot simply be associated with WAGO-4 absence from P granules. Maybe this could be made clearer throughout the text.

We agree that the fertility phenotype observed in the 3xFG mutants is likely multifactorial. We examined the localization of several factors, but of course could not check all germ granule-localized factors. In addition to WAGO-4, we detected mislocalization of GLH-4, and it is possible that other granule proteins are also affected, although not captured by our current analyses. While our data support a significant role for WAGO-4 mislocalization, we also note in the Results section (Line 273) that the unfertilized oocyte phenotype is more pronounced in 3xFG mutants than in WAGO-4 mutants:

“Notably, the frequency of unfertilized oocytes at 26°C was higher in 3×FG mutants compared to all wago-4 mutants. While around 80% of 3×FG mutants contained unfertilized oocytes, only around 50% of worms in the wago-4 mutant strains showed this phenotype (Fig. 3H). This suggests that additional factors may contribute to the more severe phenotype in 3×FG mutants. Nevertheless, our results indicate that reduced WAGO-4 partitioning in the germ granules is likely a major contributor to the fertility-related phenotype observed in 3×FG mutants.”

In line with this point, we have now clarified this interpretation more directly in the revised manuscript (Line 294).

“Together, these findings demonstrate that wago-4 mutants and 3×FG mutants exhibit similar germline phenotypes and that the FG mutations in GLH proteins and WAGO-4 mislocalization likely impair fertility through the same pathway. We cannot entirely exclude unintended gain-of-function effects resulting from WAGO-4 mislocalization to ectopic sites (cytoplasm, mitochondria, plasma membrane) and we acknowledge that GLH-4 and potentially other granule proteins are also mislocalized in the 3×FG mutant, which could lead to a complex phenotype. However, the consistency of phenotypes across the diverse WAGO-4 mislocalization contexts and the loss-of-function mutant, and similarity with 3×FG mutant phenotypes suggest that impaired WAGO-4 granule localization is likely the primary contributor to the observed fertility defects. This indicates that WAGO-4 function is compromised when its normal enrichment in germ granules is disrupted.”

Referee #3:

Germ granules are biomolecular condensates that play critical roles in germ cell development and fidelity through small RNA regulatory pathways. Argonautes are critical players within germ granules and these RNA regulatory pathways, but the connection between proper granule formation and Argonaute function remains an important biological question. Here, Jelenic, et al. study the requirement of intrinsically disordered regions for Argonaute WAGO-4 assembly and function in P granules. Mutation and/or deletion of the FG repeats of the GLH Vasa helicases lead to decreased fecundity at higher temperatures, despite apparent germ granule formation. PGL-3 BioID identified association with WAGO-4 protein, and note that the GLH FG mutants (3×FG) cause mislocalization of WAGO-4 from P and Z granules. This mislocalization was also modestly observed in an in vitro granule assembly assay. Modification of WAGO-4 to mislocalize it from germ granules also affected fecundity and caused a mortal germline phenotype. mRNA-seq in mutant worm strains revealed differences between the mutants and WAGO-4 knockout worms. WAGO-4 small RNA-seq showed evidence that the Argonaute loses the robust ability to bind to 22G-RNAs antisense to WAGO-4 specific, non- CSR-1 Argonaute, RNA targets. The manuscript proposes a model where mislocalization from germ granules changes the loading dynamics of WAGO-4.

This manuscript is clearly written and well cited. I appreciate that it potentially (see Major comments) identifies a mutant protein responsible for a specific phenotype, rather than a null mutation that disrupts the entire system. However, the genomic findings are not entirely definitive, potentially due to the delicate molecular balance between CSR-1 and WAGO-4 that may result in only subtle effects. The study would benefit from some

insight into how WAGO-4's molecular function might be impacted by its mislocalization. Nonetheless, the work builds on previous studies and will be of interest to researchers focused on granule structure and function, Argonaute proteins, and RNA and developmental biology more generally.

We would like to thank Referee #3 for their positive comments about the overall clarity and significance of our study and for their valuable suggestions on improving the work. We have addressed the queries in the revised manuscript as described below.

MAJOR

1. Figure 1B; 2F; S1C,E,J: Recommend reporting complete statistics between a and b groups.
2. Figure 1D; 2G; S1H,I: Recommend reporting statistics between strains and categories.
3. Figure 2H: Please report statistics between N2 and mutant worms.

We appreciate the referee's suggestions regarding statistical comparisons in these figures. Because we compared multiple groups (there are up to 21 comparisons per figure), directly annotating all statistical results in the figures would make them overly crowded and difficult to evaluate and interpret. To address this while balancing figure clarity, we have provided an Excel file (Table EV6) as supplementary material containing the *p* values for all comparisons between groups and categories presented in these figures. In case of the pair-wise comparisons (like in Figure S2H, current EV2G), we reported the *p* values directly in the figures.

4. Total GLH mutant proteins levels appear different than wild type protein in Fig 2. Were total GLH protein levels assessed at different temperatures? GLH protein levels may be a cause of fecundity differences between strains.

Referees 1 and 2 had similar queries about WAGO-4 and the GLH proteins (R1, comment 4 and R2, comments 2 and 3), which we addressed above.

In short, we have examined the levels of WAGO-4 and GLH proteins in our mass spectrometry data and found that levels in the 3xFG mutants are comparable to those in wild-type animals. These data have been added to the revised manuscript (line 227):

"To evaluate whether differences in the protein levels of WAGO-4 (or GLH proteins) might contribute to the phenotype, we measured their abundance in the 3xFG mutants using mass spectrometry. Our analysis revealed that the levels of WAGO-4, as well as GLH-1, GLH-2 and GLH-4, in the 3xFG mutants are comparable to those in wild-type animals (Fig. EV2H)."

5. The mRNA- and WAGO-4 sRNA-seq assays greatly complement the mutant study, but the results are somewhat convoluted and descriptive. The manuscript will be greatly strengthened with a molecular readout assay for WAGO-4 function. Were RNAi inheritance assays attempted with the WAGO FG and mislocalization mutants (e.g. Xu et al. 2018 (DOI: 10.1016/j.celrep.2018.04.072))? The results may help further support granule localization as a site required for RNAi exogenous inheritance; if FG/localization mutant results mimics wago-4 null worms, this will support the granule as the subcellular location for WAGO-4 exogenous RNAi. I realize this moves away from the manuscript's intent on investigating WAGO-4's function in regulating endogenous genes, but it does support the model of WAGO-4 using two distinct sRNA loading pathways for exogenous and endogenous gene targets, as described in Figure 6.

We appreciate the referee's suggestion to explore RNAi inheritance assays with the 3xFG and WAGO mislocalization mutants. This is indeed an interesting approach and could provide valuable insights into the functional role of WAGO-4 granule localization. However, as the referee correctly noted, this experiment moves beyond the scope of our current study, which focuses on WAGO-4's role in regulating endogenous gene expression. We discussed this with the editor, and it was agreed that this experiment is outside the intended scope of the manuscript. We included this suggestion as a future direction in the Discussion section of the manuscript to highlight the potential of this line of research (line 548):

"Our findings indicate the importance of balancing WAGO-4 and CSR-1 activities in germline gene regulation, but the precise mechanisms underlying their interplay—whether dictated by intrinsic sRNA properties, target binding site locations, intracellular activity sites, or other factors—remain to be elucidated. Additionally, WAGO-4 localization may play a role in exogenous RNAi inheritance, as CSR-1 has not been implicated in this pathway. Although not explored in this study, granule localization could be crucial for WAGO-4 loading sRNAs involved in exogenous RNAi. This line of investigation, though beyond the scope of the current work, represents an important direction for future research."

6. Figure 6: "Our findings support a model (Fig. 6) in which WAGO-4 utilizes at least two distinct small RNA loading pathways: a cytoplasmic pathway shared with CSR-1 and a germ granule-dependent pathway required for targeting unique transcripts." This WAGO-4 sRNA loading model is not clearly depicted in the Figure.

We appreciate the referee's comment regarding the depiction of the WAGO-4 sRNA loading model in Figure 6. Because there are still unresolved details about the molecular mechanisms of the WAGO-4 and CSR-1 sRNA loading pathways, we refrained from speculating about these aspects in a more detailed model in the figure.

Instead, the current figure summarizes our main findings, which are consistent with the proposed model and reflective of the data we present. Considering the referee's comment, we decided to present this figure as a Synopsis figure rather than a definitive mechanistic model.

7. Recommend adding a statistics sentence/paragraph to the methods or statistics description in the appropriate sections to mention the software used for calculations.

We appreciate the referee's suggestion and have added a paragraph in the Methods section detailing the statistical analysis. This includes the software used for calculations and a description of the statistical tests performed (Line 903):

“Statistical analysis

Statistical analysis of large datasets (including MS, mRNA sequencing, sRNA sequencing, and bioinformatic analysis of Vasa orthologs) was performed as described in the respective sections. Gene set enrichment analysis was conducted using Fisher's exact test, implemented in RStudio (version 2024.12.0.467). To account for multiple comparisons, the Benjamini-Hochberg false discovery rate (FDR) correction was applied to the p values. To assess the enrichment of Regulated Gene Sets in temperature-related differential gene expression data, the online tool WormCat was used (Holdorf et al, 2020). All remaining statistical analyses were performed in GraphPad Prism (version 10.2.3). Statistical tests were selected based on experimental design, number of groups, sample size, data distribution, and the appropriate unit of replication. In all cases, the exact n and replication strategy are indicated in each figure legend.

For comparisons involving more than two groups, we used either the Kruskal–Wallis test or one-way ANOVA. In fecundity assays, data were frequently non-normally distributed and were analyzed using the Kruskal–Wallis test followed by Dunn's post hoc correction. Group differences were visualized using letter annotations in figures with multiple comparisons for the visual clarity; all corresponding p values are provided in Table EV6. In one experiment (Fig. 3G), 5/21 pairwise comparisons yielded uncorrected p values below 0.05, but became non-significant after Dunn's correction due to >20× p value inflation. In this case, the Benjamini–Krieger–Yekutieli (BKY) false discovery rate (FDR) correction was additionally applied (indicated in the figure) and all uncorrected, Dunn-adjusted, and FDR-adjusted values are reported in Table EV6. For phenotypic scoring (e.g., unfertilized oocyte quantification), the percentage of animals in each category was calculated per experiment, with each replicate comprising ~50 animals. Variability between replicates was low and results were consistent. Group means from three independent experiments were analyzed using one-way ANOVA. Statistical

significance was not indicated in the graphs for clarity, but corresponding *p* values are provided in Table EV6.

For pairwise comparisons, the Mann–Whitney test was used unless otherwise noted. This non-parametric test was selected due to frequent deviations from normality (assessed using D’Agostino–Pearson, Anderson–Darling, Shapiro–Wilk and Kolmogorov–Smirnov tests), or the inability to reliably assess normality in small sample sizes. One exception was PRM-MS quantification (Fig. EV3C), where Welch’s t-test was applied to log-transformed intensity ratios, following standard practice in proteomics to stabilize variance and allow fold-change interpretation.”

8. A brief description of the CRISPR-Cas strategy used should be described in the methods. Were the fluorescent and bioID genes amplified from a plasmid? It is also not clear to me whether the GLH F to A mutations were each made individually, or gene fragment blocks inserted.

We appreciate this comment and have included the description of CRISPR-Cas strategy in the Methods section, *C. elegans* strains and maintenance paragraph (Line 572):

“For CRISPR gene targeting, crRNAs, tracrRNA, Cas9, Cas12a, and HDR donor oligos were obtained from IDT. Larger insertions containing introns were amplified from plasmids (mEGFP, mScarlet-1, and TurboID– information in the Reagents and Tools table) or from the C. elegans genomic DNA (MTS and PH– sequences provided in Table EV5) using PCR with primers that incorporated 35 to 50 nucleotide homology arms. Donor constructs for glh-1 and glh-4 containing F→A mutations (including 145 bp or 420 bp homology arms) were synthesized by Genewiz and amplified by PCR. All strains were verified by sequencing. The list of the strains used in this study is provided in Table EV3, and the list of gRNAs in Table EV4.”

MINOR

1. Figure 5B: The bold numbers are difficult to distinguish from the non-bold numbers. Recommend revision for clarity.

For clarity, we have additionally underlined the bold numbers to make them easier to distinguish.

2. In an ideal world, the representative images from Fig S3A,C would be presented in

the main Figure prior to the quantitation. Consider moving (a) the images to the main figure +/- splitting up the Figure to accommodate space, or (b) moving the *in vitro* data to the supplement and the germline images to the main figure as this is the more striking result.

We agree with the reviewer that, ideally, representative images and quantifications should be shown together. In the revised manuscript, we have moved the representative images from the *in vivo* experiment (previously Fig. S3A) to the main figure, positioning them before the corresponding quantification (now Fig. 3A and B). To maintain clarity and focus in the main figure, we kept the *in vitro* representative images (previously Fig. S3C) in the supplementary figure (now Fig. EV3A).

Dear Dr. Jelenic,

Thank you for submitting your manuscript for consideration by the EMBO Journal. It has now been seen by three referees whose comments are enclosed. As you will see, all three referees are now broadly in favour of publication. The only remaining issue are several small things detected by our editorial assistance team. I am sure you can easily fix these (please see more details below).

Given the referees' positive recommendations, I am happy to inform you that your paper is accepted in principle, pending that these very minor issues are addressed.

We generally allow three months as standard revision time. Yet, I am sure you can submit your revision much sooner.

Thank you for the opportunity to consider your work for publication. I look forward to your final revision so we can formally accept it.

Yours sincerely,

Yehu Moran
Academic Editor
The EMBO Journal

We realize that it is difficult to revise to a specific deadline. In the interest of protecting the conceptual advance provided by the work, we recommend a revision within 3 months (20th Nov 2025). Please discuss the revision progress ahead of this time with the editor if you require more time to complete the revisions.

specific comments by the editorial assistance team

*Author contributions: please remove from manuscript text and keep it only in our submission system.

*DATASET EV LEGENDS: Please rename Table EV2 and make it "Dataset EV1". Please adjust the numbering of the subsequent tables accordingly and ensure that the title is included in the legends of each table.

*APPENDIX 1 FILE WITH Table of Contents: only contains supplementary methods. Can these be added to Methods in the main manuscript text? We believe this can be easier for the readers and we do not have a limitation on length.

*REAGENT TABLE: please remove from the manuscript text and upload the table as a separate file and adjust the EV table numbering accordingly.

*SYNOPSIS IMAGE: included, but please resize to 550 pixels wide x 300 - 600 pixels high.

- Data Accessibility Statement: Please remember to make sure the data will become publicly available before the paper is published. This is your responsibility as an author.

- Figure legends:

1. Please note that the exact p values are not provided in the legends of figures 1B, 3B, D, G; EV1 B, C, D, E, G, L. Please correct.

2. Please indicate the statistical test used for data analysis in the legends of figures 2D, EV4 D, EV5 B.

3. Please note that information related to n is missing in the legends of figures 1B, 3G, EV1 B, C, D, E, G, H, L; EV3 B, EV4B, C, D, F, G. This should be fixed.

specific referee comments

Referee #1:

The authors addressed all the concerns and I will be happy to see the paper published.

Referee #2:

The authors have addressed all points that I had raised on the previous version of their manuscript. They have adequately responded in their rebuttal letter, clarified unclear issues and therefore I am satisfied with the response to my comments.

Referee #3:

This reviewer thanks the authors for their thoughtful revisions, considerations, and comments of my previous report. My concerns for the manuscript have been addressed in this resubmission.

The authors addressed the remaining editorial issues.

Dear Dr. Jelenic,

I am pleased to inform you that your manuscript has been now formally accepted for publication in the EMBO Journal.

Yours sincerely,

Yehu Moran
Academic Editor
The EMBO Journal
